# Recurrent Memory for Online Interdomain Gaussian Processes

**Wenlong Chen**[1,*], **Naoki Kiyohara**[1,2,*], **Harrison Bo Hua Zhu**[3,1,*],
**Jacob Curran-Sebastian**[3], **Samir Bhatt**[3,1], **Yingzhen Li**[1],
[1]Imperial College London  [2]Canon Inc.  [3]University of Copenhagen
wenlong.chen21@imperial.ac.uk  n.kiyohara23@imperial.ac.uk
harrison.zhu@sund.ku.dk  yingzhen.li@imperial.ac.uk

## Abstract

We propose a novel online Gaussian process (GP) model that is capable of capturing long-term memory in sequential data in an online learning setting. Our model, Online HiPPO Sparse Variational Gaussian Process (OHSVGP), leverages the HiPPO (High-order Polynomial Projection Operators) framework, which is popularized in the RNN domain due to its long-range memory modeling capabilities. We interpret the HiPPO time-varying orthogonal projections as inducing variables with time-dependent orthogonal polynomial basis functions, which allows the SVGP inducing variables to memorize the process history. We show that the HiPPO framework fits naturally into the interdomain GP framework and demonstrate that the kernel matrices can also be updated online in a recurrence form based on the ODE evolution of HiPPO. We evaluate OHSVGP with online prediction for 1D time series, continual learning in discriminative GP model for data with multidimensional inputs, and deep generative modeling with sparse Gaussian process variational autoencoder, showing that it outperforms existing online GP methods in terms of predictive performance, long-term memory preservation, and computational efficiency.

## 1 Introduction

Gaussian processes (GPs) are popular choices for modeling time series due to their functional expressiveness and uncertainty quantification abilities [Roberts et al., 2013, Fortuin et al., 2020]. However, GPs are computationally expensive and memory intensive, with cubic and quadratic complexities, respectively. In online regression settings, such as weather modeling, the number of time steps can be very large, quickly making GPs infeasible. Although variational approximations, such as utilizing sparse inducing points (SGPR [Titsias, 2009]; SVGP [Hensman et al., 2013, 2015a]) and Markovian GPs [Särkkä and Solin, 2019, Wilkinson et al., 2021], have been proposed to address the computational complexity, it would still be prohibitive to re-fit the GP model from scratch every time new data arrives. Bui et al. [2017] proposed an online sparse variational GP (OSVGP) learning method that sequentially updates the GP posterior distribution only based on the newly arrived data. However, as indicated in their paper, their models may not maintain the memory of the previous data, as the inducing points will inevitably shift as new data arrive. This is a major drawback, as their models may not model long-term memory unless using a growing number of inducing points.

In deep learning, as an alternative to Transformers [Vaswani et al., 2017], significant works on state space models (SSMs) have been proposed to model long-term memory in sequential data. Originally proposed to instill long-term memory in recurrent neural networks, the HiPPO (High-order Polynomial Projection Operators) framework [Gu et al., 2020] provides mathematical foundations for

---

*Equal contribution.
Source Code: `https://github.com/harrisonzhu508/HIPPOSVGP`.

compressing continuous-time signals into memory states through orthogonal polynomial projections. HiPPO is computationally efficient and exhibits strong performance in long-range memory tasks, and forms the basis for the state-of-the-art SSMs, e.g., structured state space sequential (S4) model [Gu et al., 2022] and Mamba [Gu and Dao, 2023, Dao and Gu, 2024].

Inspired by HiPPO, we propose Online HiPPO SVGP (OHSVGP), by applying the HiPPO framework to SVGP in order to leverage the long-range memory modeling capabilities. Our method interprets the HiPPO time-varying orthogonal projections as inducing variables of an interdomain SVGP [Lázaro-Gredilla and Figueiras-Vidal, 2009, Leibfried et al., 2020, Van der Wilk et al., 2020], where the basis functions are time-dependent orthogonal polynomials. We show that we are able to significantly resolve the memory-loss issue in OSVGP, thereby opening up the possibility of applying GPs to long-term online learning tasks. In summary, our contributions include:

- (Section 3) We demonstrate that HiPPO integrates into the interdomain GPs by interpreting the HiPPO projections as inducing variables with time-dependent orthogonal polynomial basis functions. This allows the inducing variables to compress historical data, capturing long-term information.
- (Section 3.2 & 5.1) We show that the kernel matrices can leverage the efficient ODE evolution of the HiPPO framework, bringing an extra layer of computational efficiency to OHSVGP.
- (Section 5) We demonstrate OHSVGP on a variety of online/continual learning tasks including time series prediction, continual learning on UCI benchmarks, and continual learning in Gaussian process variational autoencoder, showing that it outperforms other online sparse GP baselines in terms of predictive performance, long-term memory preservation, and computational efficiency.

## 2   Background

In this section, we provide a brief overview of GPs, inducing point methods, online learning with GPs, and Gaussian process variational autoencoders. In addition, we review the HiPPO method, which is the basis of our proposed method.

### 2.1   Gaussian processes

Let $\mathcal{X}$ be the input space. For time series data, $\mathcal{X} = [0, \infty)$, the set of non-negative real numbers. A Gaussian process (GP) $f \sim \mathcal{GP}(0, k)$ is defined with a covariance function $k : \mathcal{X} \times \mathcal{X} \to \mathbb{R}$. It has the property that for any finite set of input points $\mathbf{X} = [x_1, \ldots, x_n]^{\mathsf{T}}$, the random vector $\mathbf{f} \equiv f(\mathbf{X}) = [f(x_1), \ldots, f(x_n)]^{\mathsf{T}} \sim \mathcal{N}(0, \mathbf{K_{ff}})$, where $\mathbf{K_{ff}}$ is the kernel matrix with entries $[k(\mathbf{X}, \mathbf{X})]_{ij} \equiv [\mathbf{K_{ff}}]_{ij} = k(x_i, x_j)$. For notational convenience and different sets of input points $\mathbf{X}_1$ and $\mathbf{X}_2$, we denote the kernel matrix as $\mathbf{K_{f_1 f_2}}$ or $k(\mathbf{X}_1, \mathbf{X}_2)$. The computational and memory complexities of obtaining the GP posterior on $\mathbf{X}_1$ scale cubically and quadratically respectively, according to $n_1 = |\mathbf{X}_1|$. Given responses $\mathbf{y}$ and inputs $\mathbf{X}$, a probabilistic model can be defined as $y_i \sim p(y_i \mid f(x_i))$ with a GP prior $f \sim \mathcal{GP}(0, k)$, where $p(y_i \mid f(x_i))$ is the likelihood distribution. However, for non-conjugate likelihoods, the posterior distribution is intractable, and approximate inference methods are required, such as, but not limited to, variational inference [Titsias, 2009, Hensman et al., 2013, 2015a] and Markov chain Monte Carlo (MCMC) [Hensman et al., 2015b].

### 2.2   Variational inference and interdomain Gaussian processes

To address the intractability and cubic complexity of GPs, Sparse Variational Gaussian Processes (SVGP; [Titsias, 2009, Hensman et al., 2013, 2015b]) cast the problem as an optimization problem. By introducing $M$ inducing points $\mathbf{Z} \in \mathcal{X}^M$ that correspond to $M$ inducing variables $\mathbf{u} = [f(\mathbf{z}_1), \ldots, f(\mathbf{z}_M)]^{\mathsf{T}}$, the variational distribution $q(\mathbf{f}, \mathbf{u})$ is defined as $q(\mathbf{f}, \mathbf{u}) := p(\mathbf{f} \mid \mathbf{u}) q_\theta(\mathbf{u})$, where $q_\theta(\mathbf{u})$ is the variational distribution of the inducing variables with parameters $\theta$. Then, the evidence lower bound (ELBO) is defined as

$$\log p(\mathbf{y}) \geq \sum_{i=1}^n \mathbb{E}_{q(f_i)}[\log p(y_i \mid f_i)] - \mathrm{KL}\left[q_\theta(\mathbf{u}) \| p(\mathbf{u})\right] =: \mathcal{L}_\theta, \tag{1}$$

where $q(f_i) = \int p(f_i \mid \mathbf{u}) q_\theta(\mathbf{u}) \mathrm{d}\mathbf{u}$ is the posterior distribution of $f_i \equiv f(x_i)$. Typical choices for the variational distribution are $q_\theta(\mathbf{u}) = \mathcal{N}(\mathbf{u}; \mathbf{m_u}, \mathbf{S_u})$, where $\mathbf{m_u}$ and $\mathbf{S_u}$ are the free-form mean

and covariance of the inducing variables, and yields the posterior distribution:

$$q(f_i) = \mathcal{N}(f_i; \mathbf{K}_{f_i \mathbf{u}} \mathbf{K}_{\mathbf{uu}}^{-1} \mathbf{m_u}, \mathbf{K}_{f_i f_i} - \mathbf{K}_{f_i \mathbf{u}} \mathbf{K}_{\mathbf{uu}}^{-1} [\mathbf{K_{uu}} - \mathbf{S_u}] \mathbf{K}_{\mathbf{uu}}^{-1} \mathbf{K}_{\mathbf{u} f_i}). \tag{2}$$

When the likelihood is conjugate Gaussian, the ELBO can be optimized in closed form and $\mathbf{m_u}$ and $\mathbf{S_u}$ can be obtained in closed form (SGPR; Titsias [2009]). In addition to setting the inducing variables as the function values, interdomain GPs [Lázaro-Gredilla and Figueiras-Vidal, 2009] propose to generalize the inducing variables to $u_m := \int f(x) \phi_m(x) \mathrm{d}x$, where $\phi_m(x)$ are basis functions, to allow for further flexibility. This yields $[\mathbf{K}_{f\mathbf{u}}]_m = \int k(x, x') \phi_m(x') \mathrm{d}x'$ and $[\mathbf{K_{uu}}]_{nm} = \iint k(x, x') \phi_n(x) \phi_m(x') \mathrm{d}x \mathrm{d}x'$.

We see that the interdomain SVGP bypasses the selection of the inducing points $\mathbf{Z} \in \mathbb{R}^M$, and reformulates it with the selection of the basis functions $\phi_i$. The basis functions dictate the structure of the kernel matrices, which in turn modulate the function space of the GP approximation. In contrast, SVGP relies on the inducing points $\mathbf{Z}$, which can shift locations according to the training data. Some examples of basis functions include Fourier basis functions [Hensman et al., 2018] and the Dirac delta function $\delta_{\mathbf{z}_m}$, the latter recovering the standard SVGP inducing variables.

## 2.3 Online Gaussian processes

In this paper, we focus on online learning with GPs, where data arrives sequentially in batches $(\mathbf{X}_{t_1}, \mathbf{y}_{t_1}), (\mathbf{X}_{t_2}, \mathbf{y}_{t_2}), \ldots$ etc. For example, in the time series prediction setting, the data arrives in intervals of $(0, t_1), (t_1, t_2), \ldots$ etc. The online GP learning problem is to sequentially update the GP posterior distribution as data arrives. Suppose that we have already obtained $p_{t_1}(y|f) p_{t_1}(f|\mathbf{u}_{t_1}) q_{t_1}(\mathbf{u}_{t_1})$ of the likelihood and variational approximation (with inducing points $\mathbf{Z}_{t_1}$), from the first data batch $(\mathbf{X}_{t_1}, \mathbf{y}_{t_1})$. Online SVGP (OSVGP; [Bui et al., 2017]) utilizes the online learning ELBO

$$\sum_{i=1}^{n_{t_2}} \mathbb{E}_{q_{t_2}(f_i)} \left[ \log p_{t_2}(y_i \mid f_i) \right] + \mathrm{KL} \left( \tilde{q}_{t_2}(\mathbf{u}_{t_1}) \,\|\, p_{t_1}(\mathbf{u}_{t_1}) \right) \\ - \mathrm{KL} \left( \tilde{q}_{t_2}(\mathbf{u}_{t_1}) \,\|\, q_{t_1}(\mathbf{u}_{t_1}) \right) - \mathrm{KL} \left( q_{t_2}(\mathbf{u}_{t_2}) \,\|\, p_{t_2}(\mathbf{u}_{t_2}) \right), \tag{3}$$

where $y_i \in \mathbf{y}_{t_2}$ for $i = 1, \ldots, n_{t_2}$ and $\tilde{q}_{t_2}(\mathbf{u}_{t_1}) := \int p_{t_2}(\mathbf{u}_{t_1}|\mathbf{u}_{t_2}) q_{t_2}(\mathbf{u}_{t_2}) \mathrm{d}\mathbf{u}_{t_2}$. Unfortunately, with more and more tasks, OSVGP may not capture the long-term memory in the data since as new data arrives, it is not guaranteed that the inducing points after optimization can sufficiently cover all the previous tasks' data domains.

## 2.4 Gaussian process variational autoencoders

Gaussian processes can be embedded within a variational autoencoder (VAE; [Kingma and Welling, 2014]) framework, giving rise to the Gaussian process variational autoencoder (GPVAE; [Casale et al., 2018, Fortuin et al., 2020, Ashman et al., 2020, Jazbec et al., 2021, Zhu et al., 2023]). For sparse GPs with inducing variables, Jazbec et al. [2021] introduced the SVGPVAE, which combines the sparse variational GP (SVGP) with the VAE formulation. The likelihood $p(y \mid \varphi_\theta(f))$ is parameterized by a decoder network $\varphi_\theta$, which takes GP latent draws $f$ as input, together with the variational inducing posterior $q_\theta(\mathbf{u} \mid y)$. This posterior, $q_\theta(\mathbf{u} \mid \phi(y))$, is parameterized by the encoder network $\phi$. Finally, the latent GP $f$ is typically modeled as a multi-output GP with independent components. GPVAEs have been shown to successfully model high-dimensional time series such as weather data and videos [Zhu et al., 2023, Fortuin et al., 2020]. In this work, we consider the SVGPVAE model defined in Jazbec et al. [2021] for one set of our experiments, and the detailed specification of the model and training objective can be found in Appendix D.2.

## 2.5 HiPPO: recurrent memory with optimal polynomial projections

The HiPPO framework [Gu et al., 2020] provides mathematical foundations for compressing continuous-time signals into finite-dimensional memory states through optimal polynomial projections. Given a time series $y(t)$, HiPPO maintains a memory state $c(t) \in \mathbb{R}^M$ that optimally approximates the historical signal $\{y(x)\}_{x \leq t}$. The framework consists of a time-dependent measure $\omega^{(t)}(x)$ over $(-\infty, t]$ that defines input importance, along with normalized polynomial basis functions $\{g_n^{(t)}(x)\}_{n=0}^{M-1}$ that are orthonormal under $\omega^{(t)}(x)$, satisfying $\int_{-\infty}^{t} g_m^{(t)}(x) g_n^{(t)}(x) \omega^{(t)}(x) \mathrm{d}x =$

$\delta_{mn}$. The historical signal is encoded through projection coefficients given by $c_n(t) = \int_{-\infty}^{t} y(x) g_n^{(t)}(x) \omega^{(t)}(x) \mathrm{d}x$. This yields the approximation $y(x) \approx \sum_{n=0}^{M-1} c_n(t) g_n(x)$ for $x \in (-\infty, t]$, minimizing the $L^2$-error $\int_{-\infty}^{t} \|y(x) - \sum_n c_n(t) g_n(x)\|^2 \omega^{(t)}(x) \mathrm{d}x$. Differentiating $\mathbf{c}(t) := [c_0(t), \ldots, c_{M-1}(t)]^\intercal$ induces a linear ordinary differential equation $\frac{\mathrm{d}}{\mathrm{d}t} \mathbf{c}(t) = \mathbf{A}(t) \mathbf{c}(t) + \mathbf{B}(t) y(t)$ with matrices $\mathbf{A}(t), \mathbf{B}(t)$ encoding measure-basis dynamics. Discretization yields the recurrence of the form $\mathbf{c}_t = \mathbf{A}_t \mathbf{c}_{t-1} + \mathbf{B}_t y_t$ enabling online updates. The structured state space sequential (S4) model [Gu et al., 2022] extends HiPPO with trainable parameters and convolutional kernels, while Mamba [Gu and Dao, 2023, Dao and Gu, 2024] introduces hardware-aware selective state mechanisms, both leveraging HiPPO for efficient long-range memory modeling. HiPPO supports various measure-basis configurations [Gu et al., 2020, 2023]. A canonical instantiation, HiPPO-LegS, uses a uniform measure $\omega^{(t)}(x) = \frac{1}{t} \mathbf{1}_{[0,t]}(x)$ with scaled Legendre polynomials adapted to $[0, t]$, $g_n^{(t)}(x) = (2m+1)^{1/2} P_m\left(\frac{2x}{t} - 1\right)$. This uniform measure encourages HiPPO-LegS to keep the whole past in memory.

# 3 Interdomain inducing point Gaussian processes with HiPPO

We bridge the HiPPO framework with interdomain Gaussian processes by interpreting HiPPO's state vector defined by time-varying orthogonal projections as interdomain inducing points. This enables adaptive compression of the history of a GP while preserving long-term memory.

## 3.1 HiPPO as interdomain inducing variables

Recall that in an interdomain setting in Section 2.2, inducing variables are defined through an integral transform against a set of basis functions. Let $f \sim \mathcal{GP}(0, k)$, and consider time-dependent basis functions $\phi_m^{(t)}(x) = g_m^{(t)}(x) \omega^{(t)}(x)$, where $g_m^{(t)}$ are the orthogonal functions of HiPPO and $\omega^{(t)}$ is the associated measure. We define the corresponding interdomain inducing variables as $u_m^{(t)} = \int f(x) \phi_m^{(t)}(x) \mathrm{d}x$, which is not a standard random variable as in Section 2.2. Rather, it is a random functions (i.e. stochastic processes) over time ($u_m^{(t)} \equiv u_m(t)$) due to time-dependent basis functions. These inducing variables adapt in time, capturing long-range historical information in a compact form via HiPPO's principled polynomial projections.

## 3.2 Adapting the kernel matrices over time

When new observations arrive at later times in a streaming scenario, we must adapt both the prior cross-covariance $\mathbf{K_{fu}}$ and the prior covariance of the inducing variables $\mathbf{K_{uu}}$. In particular, the basis functions in our HiPPO construction evolve with time, so the corresponding kernel quantities also require updates. Below, we describe how to compute and update these matrices at a new time $t_2$ given their values at time $t_1$. For clarity, we first discuss $\mathbf{K_{fu}}$, then $\mathbf{K_{uu}}$.

**Prior cross-covariance $\mathbf{K_{fu}^{(t)}}$.** Recall that for a single input $x_n$, the prior cross-covariance with the $m$-th inducing variable is $\left[\mathbf{K_{fu}^{(t)}}\right]_{nm} = \int k(x_n, x) \phi_m^{(t)}(x) \mathrm{d}x$. We can compute the temporal evolution of $\mathbf{K_{fu}^{(t)}}$ in a manner consistent with the HiPPO approach, leveraging the same parameters $\mathbf{A}(t)$ and $\mathbf{B}(t)$. Specifically,

$$\frac{\mathrm{d}}{\mathrm{d}t} \left[\mathbf{K_{fu}^{(t)}}\right]_{n,:} = \mathbf{A}(t) \left[\mathbf{K_{fu}^{(t)}}\right]_{n,:} + \mathbf{B}(t) k(x_n, t), \tag{4}$$

where $\left[\mathbf{K_{fu}^{(t)}}\right]_{n,:}$ is the $n$-th row of $\mathbf{K_{fu}^{(t)}}$. The matrices $\mathbf{A}(t)$ and $\mathbf{B}(t)$ depend on the specific choice of the HiPPO measure and basis functions. In our experiments, we employ HiPPO-LegS, whose explicit matrix forms are provided in Appendix A. One then discretizes in $t$ (e.g. using an Euler method or a bilinear transform) to obtain a recurrence update rule.

**Prior inducing covariance $\mathbf{K_{uu}^{(t)}}$.** The $\ell m$-th element of the prior covariance matrix for the inducing variables is given by $\left[\mathbf{K_{uu}^{(t)}}\right]_{\ell m} = \iint k(x, x') \phi_\ell^{(t)}(x) \phi_m^{(t)}(x') \mathrm{d}x \mathrm{d}x'$. Since $k(x, x')$ depends on both

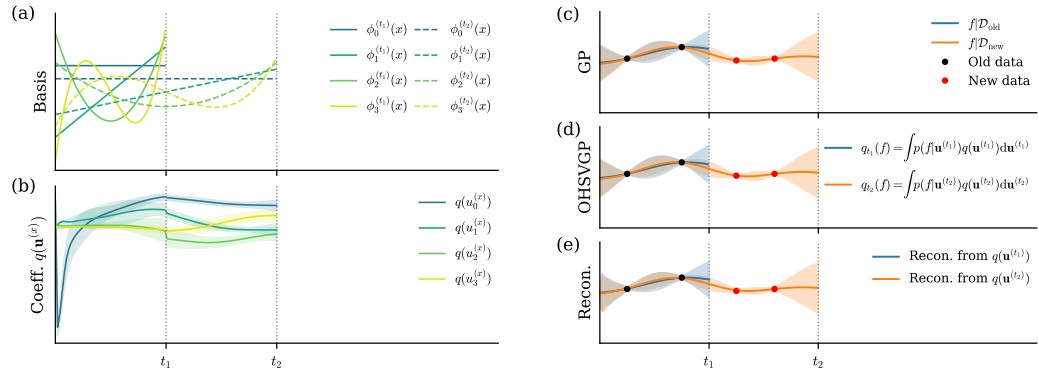

Figure 1: Online HiPPO Sparse Variational Gaussian Process (OHSVGP) on a toy time series with 2 tasks. Here $x$ is used to denote arbitrary time index. **(a)** Time-dependent basis functions with end time index $x = t_1$ and $x = t_2$. **(b)** Evolution of optimal approximate posterior of inducing variables (mean $\pm 2$ marginal standard deviation). **(c)**, **(d)**, **(e)** illustrate predictive mean $\pm 2$ standard deviation of posterior online GP, OHSVGP and finite basis reconstruction of posterior OHSVGP, respectively.

$x$ and $x'$, a recurrence update rule based on the original HiPPO formulation, which is designed for single integral, can not be obtained directly for $\mathbf{K}_{\mathbf{uu}}^{(t)}$. Fortunately, for stationary kernels, Bochner Theorem [Rudin, 1994] can be applied to factorize the double integrals into two separate single integrals, which gives rise to Random Fourier Features (RFF) approximation [Rahimi and Recht, 2007]: for a stationary kernel $k(x, x') = k(|x - x'|)$, RFF approximates it as follows: $k(x, x') \approx \frac{1}{N} \sum_{n=1}^{N} [\cos(w_n x) \cos(w_n x') + \sin(w_n x) \sin(w_n x')]$, where $w_n \sim p(w)$ is the spectral density of the kernel. Substituting this into the double integral factorizes the dependency on $x$ and $x'$, reducing $[\mathbf{K}_{\mathbf{uu}}^{(t)}]_{\ell m}$ to addition of products of one-dimensional integrals. Each integral, with the form of either $\int \cos(w_d x) \phi_\ell^{(t)}(x) \mathrm{d}x$ or $\int \sin(w_d x) \phi_\ell^{(t)}(x) \mathrm{d}x$, again corresponds to a HiPPO-ODE in time. By sampling multiple random features, updating them recurrently to time $t$, and averaging, we obtain RFF approximation of $\mathbf{K}_{\mathbf{uu}}^{(t)}$. In addition, more advanced Fourier feature approximation techniques (e.g., [Ton et al., 2018]) can be leveraged for non-stationary kernels. The details of the ODE for recurrent updates of the RFF samples appear in Appendix B.1. Alternatively, one may differentiate $\mathbf{K}_{\mathbf{uu}}^{(t)}$ directly with respect to $t$. This yields a matrix ODE of the form different from the original HiPPO formulation. For details, see Appendix B.2. Empirically, a vanilla implementation of this approach shows numerical unstability. Hence, we conduct our experiments based on RFF approximation.

**Sequential variational updates.** Having obtained $\mathbf{K}_{\mathbf{fu}}^{(t_2)}, \mathbf{K}_{\mathbf{uu}}^{(t_2)}$ at a new time $t_2 > t_1$, we perform variational updates following the online GP framework described in Section 2.3. This ensures the posterior at time $t_2$ remains consistent with both the new data and the previous posterior at time $t_1$, based on $\mathbf{K}_{\mathbf{fu}}^{(t_1)}, \mathbf{K}_{\mathbf{uu}}^{(t_1)}$. Overall, this procedure endows interdomain HiPPO-based GPs with the ability to capture long-term memory online. By viewing the induced kernel transforms as ODEs in time, we efficiently preserve the memory of past observations while adapting our variational posterior in an online fashion. Figure 1b illustrates the evolution of the optimal posterior $q(\mathbf{u}^{(x)})$ as time $x$ increases on a toy online time series regression problem with two tasks, where $x$ determines the end of the recurrent update for the prior cross and inducing covariance matrices (evolved up to $\mathbf{K}_{\mathbf{fu}}^{(x)}$ and $\mathbf{K}_{\mathbf{uu}}^{(x)}$, respectively). Furthermore, when $x > t_1$, we will update $q(\mathbf{u}^{(x)})$ online with the two data points from the second task by optimizing the online ELBO (Eq. 3), which gives the discrete jump at $x = t_1$. Figure 1d shows the posterior OHSVGP compared with the fit of the gold-standard online GP in Figure 1c. Notably, if $f \sim q_t(f)$, then $q(u_m^{(t)}) \stackrel{d}{=} \int f(x) \phi_m^{(t)}(x) \mathrm{d}x$ (detailed derivation in Appendix C). Therefore, our framework also provides a finite basis approximation of the posterior OHSVGP as a byproduct: $f = \sum_{m=1}^{M} u_m^{(t)} g_m^{(t)}(x), u_m^{(t)} \sim q(u_m^{(t)})$. Figure 1e plots the finite basis approximation/reconstruction and it is close to the posterior OHSVGP for this simple example.

### 3.3 Extending OHSVGP to multidimensional input

For multidimensional input data, suppose there is a time order for the first batch of training points with inputs $\{\boldsymbol{x}_n^{(1)}\}_{n=1}^{N_1}$, such that $\boldsymbol{x}_i^{(1)}$ appears after $\boldsymbol{x}_j^{(1)}$ if $i > j$, and we further assume $\boldsymbol{x}_i$ appears at time index $i\Delta t$ (i.e., $\mathbf{x}(i\Delta t) = \mathbf{x}_i^{(1)}$), where $\Delta t$ is a user-specified constant step size. In this case, we can again obtain interdomain prior covariance matrices via HiPPO recurrence. For example, a forward Euler method applied to the ODE in Eq. 4 for $\mathbf{K}_{\mathbf{fu}}^t$ yields

$$[\mathbf{K}_{\mathbf{fu}}^{((i+1)\Delta t)}]_{n,:} = [\mathbf{I} + \Delta t \mathbf{A}(i\Delta t)][\mathbf{K}_{\mathbf{fu}}^{(i\Delta t)}]_{n,:} + \Delta t \mathbf{B}(i\Delta t)k\left(\mathbf{x}_n^{(1)}, \mathbf{x}_i^{(1)}\right). \tag{5}$$

The equation above can be viewed as a discretization (with step size $\Delta t$) of an ODE solving path integrals of the form $\int_0^{N_1 \Delta t} k\left(\boldsymbol{x}_n^{(1)}, \boldsymbol{x}(s)\right) \phi_m^{(t)}(s)ds$. The $i$-th training input $\boldsymbol{x}_i^{(1)}$ is assumed to be $\boldsymbol{x}_i^{(1)} := \boldsymbol{x}(i\Delta t)$ and thus the path integral is approximately solved with discretized recurrence based on the training inputs corresponding to $\{\boldsymbol{x}(i\Delta t)\}_{i=1}^N$. We continue the recurrence for the second task with ordered training inputs $\{\boldsymbol{x}_n^{(2)}\}_{n=1}^{N_2}$ by assigning time index $(N_1 + i)\Delta t$ to its $i$-th instance. and keep the recurrence until we learn all the tasks continually. In practice, one may use a multiple of $\Delta t$ as the step size to accelerate the recurrence, e.g., instead of using all the training inputs, one can compute the recurrence based on $\{\boldsymbol{x}_1, \boldsymbol{x}_3, \boldsymbol{x}_5, \cdots\}$ only by using step size $2\Delta t$. When there is no natural time order for training instances in each task, such as in standard continual learning applications, we need to sort the instances with some criterion to create pseudo time order to fit OHSVGP, similar to the practice of applying SSMs to non-sequence data modalities, e.g., SSMs, when applied to vision tasks, assign order to patches in an image for recurrence update of the memory [Zhu et al., 2024]. In our experiments, we show that the performance of OHSVGP, when applied to continual learning, depends on the sorting criterion used.

## 4 Related work

**Online sparse GPs.** Previous works mainly focus on reducing the sparse approximation error with different approximate inference techniques, such as variational inference [Bui et al., 2017, Maddox et al., 2021], expectation propagation [Csató and Opper, 2002, Bui et al., 2017], Laplace approximation [Maddox et al., 2021], and approximation enhanced with replay buffer [Chang et al., 2023]. The orthogonal research problem of online update of inducing points remains relatively underexplored, and pivoted-Cholesky [Burt et al., 2019] as deployed in Maddox et al. [2021], Chang et al. [2023] is one of the most effective approaches for online update of inducing points up to date. We tackle this problem by taking advantage of the long-term memory capability of HiPPO to design an interdomain inducing variable based method and the associated recurrence based online update rules. Notably, our HiPPO inducing variables in principle are compatible with all the aforementioned approximate inference frameworks since only the way of computing prior covariance matrices will be different from standard online sparse GPs.

**Interdomain GPs.** To our knowledge, OHSVGP is the first interdomain GP method in the context of online learning. Previous interdomain GPs typically construct inducing variables via integration based on a predefined measure (e.g., a uniform measure over a fixed interval [Hensman et al., 2018] or a fixed Gaussian measure [Lázaro-Gredilla and Figueiras-Vidal, 2009]) to prevent diverging covariances, and this predefined measure may not cover all regions where the time indices from future tasks are, making them unsuitable for online learning. In contrast, OHSVGP bypasses this limitation by utilizing adaptive basis functions constructed based on time-dependent measure which keeps extending to the new time region as more tasks arrive.

**Markovian GPs.** Markovian GPs [Särkkä and Solin, 2019, Wilkinson et al., 2021] have similar recurrence structure during inference and training due to their state space SDE representation. However, Markovian GPs are popularized due to their $\mathcal{O}(n)$ computational complexity and is not explicitly designed for online learning.

## 5 Experiments

**Applications & datasets.** We evaluate OHSVGP against baselines in the following tasks.

- **Time series prediction.** We consider regression benchmarks, Solar Irradiance [Lean, 2004], and Audio Signal [Bui and Turner, 2014] produced from the TIMIT database [Garifolo et al., 1993]. We preprocess the two datasets following similar procedures described in Gal and Turner [2015] and Bui et al. [2017], respectively (the train-test split is different due to random splitting). In addition, we consider a daily death-count time series from Santa Catarina State, Southern Brazil spanning the March 2020 to February 2021 COVID-19 pandemic, obtained from Hawryluk et al. [2021]. We construct online learning tasks by splitting each dataset into 10 (5 for COVID) sequential partitions with an equal number of training instances.

- **Continual learning.** We consider continual learning on two UCI datasets with multi-dim inputs, Skillcraft [Blair et al., 2013] and Powerplant [Tfekci and Kaya, 2014], using the same data preprocessing procedure as in Stanton et al. [2021]. We construct two types of continual learning problems by first sorting the data points based on either the values in their first dimension or their L2 distance from the origin, and then splitting the sorted datasets into 10 sequential tasks with an equal number of training instances.

- **High dimensional time series prediction.** We evaluate GPVAEs on hourly climate data from ERA5 [Copernicus Climate Change Service, Climate Data Store, 2023, Hersbach et al., 2023], comprising 17 variables across randomly scattered locations around the UK from January 2020 onward. The dataset is split into 10 sequential tasks of 186 hourly time steps each.

**Baseline.** We compare OHSVGP with OSVGP [Bui et al., 2017] and OVC (Online Variational Conditioning; [Maddox et al., 2021]). At the beginning of each task, OSVGP initialize the induing points by sampling from the old inducing points and the new data points, while OVC initializes them via pivoted-Cholesky [Burt et al., 2019] and we consider both fixing the initialized inducing points as in Chang et al. [2023] (OVC) or keep training them as in Maddox et al. [2021] (OVC-optZ). For time series regression with Gaussian likelihood, we consider OHSGR and OSGPR (OHSVGP and OSVGP based on closed form ELBO), and we further consider OVFF (OSGPR based on variational Fourier feature (VFF), an interdomain inducing point approach from Hensman et al. [2018]).

**Hyperparameters.** Within each set of experiments, all the models are trained using Adam [Kingma and Ba, 2015] with the same learning rate and number of iterations. For OHSVGP, we construct inducing variables based on HiPPO-LegS [Gu et al., 2020] (see Appendix F.4 for visualizations of using other HiPPO variants) and use 1000 RFF samples. We use ARD-RBF kernel, except for OVFF, tailored specifically to Matérn kernels, where we use Matérn-$\frac{5}{2}$ kernel instead. Similar to Maddox et al. [2021], we do not observe performance gain by keeping updating kernel hyperparameters online, and we include results with trainable kernel hyperparameters in Appendix F.2 for time series regression, but the performance becomes unstable when number of tasks is large. Thus, we either only train the kernel hyperparameters during the initial task and keep them fixed thereafter (Section 5.3) or obtain them from a full GP model trained over the initial task. It is also worth noting that OVFF requires computing covariances as integrals over a predefined interval covering the whole range of the time indices from all tasks (including unobserved ones), which is impractical in real online learning scenarios. For our experiments, we set the two edges of this interval to be the minimum and maximum time index among the data points from all the tasks, respectively.

**Evaluations & metrics.** We report results in Negative Log Predictive Density (NLPD) in the main text, and Root Mean Squared Error (RMSE) in Appendix F.1 (expected calibration error (ECE; [Guo et al., 2017]) for COVID data instead), which shows consistent conclusions as NLPD. We report the mean and the associated 95% confidence interval obtained from 5 (3 for experiments on ERA5) independent runs.

## 5.1 Online time series prediction

**Time series regression.** Figure 2 shows NLPD (over the past tasks) of different methods during online learning through the 10 tasks for Solar Irradiance and Audio dataset. Overall, OHSGPR consistently achieves the best performance with OVC performing competitively, especially as we learn more and more tasks, suggesting OHSGPR effectively preserves long-term memory through its HiPPO-based memory mechanism. OSGPR shows catastrophic forgetting starting around task

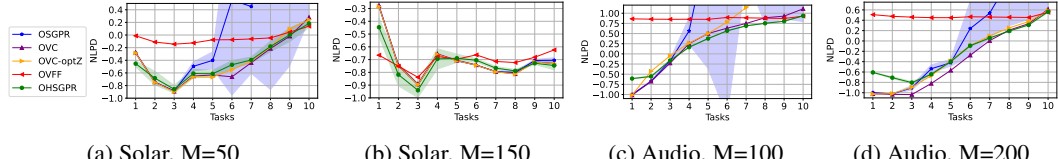

|   (a) Solar, M=50   |   (b) Solar, M=150   |   (c) Audio, M=100   |   (d) Audio, M=200   |

Figure 2: Test set NLPD over the learned tasks vs. number of learned tasks for Solar Irradiance and Audio signal prediction dataset.

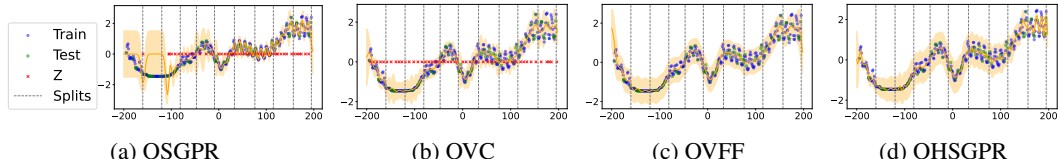

|   (a) OSGPR   |   (b) OVC   |   (c) OVFF   |   (d) OHSGPR   |

Figure 3: Predictive mean $\pm 2$ standard deviation of OSGPR, OVC, OVFF, and OHSGPR after task 10 of the Solar dataset. M = 50 inducing variables are used.

5, especially when the number of inducing points $M$ is small. Although OVC-optZ also initializes inducing points with pivoted-Cholesky as OVC, with further optimization, its performance starts to degrade starting from task 6 for the audio dataset when $M = 100$, which suggests the online ELBO objective cannot guarantee optimal online update of inducing points that preserve memory. OVFF tends to perform well at the later stage. However, during the first few tasks, it underfits the data significantly compared with other methods since its inducing variables are computed via integration over a predefined interval capturing the region of all the tasks, which is unnecessarily long and suboptimal for learning at the early stage.

In Figure 3, we compare the final predictive distributions for different methods after finishing online learning all 10 tasks of Solar Irradiance. The inducing points $\mathbf{Z}$ for OSGPR tend to move to the regions where the later tasks live after online training, and the prediction of OSGPR in the initial regions without sufficient inducing points becomes close to the uninformative prior GP. In contrast, OHSGPR maintains consistent performance across both early and recent time periods.

**Infectious disease modeling** We replace the Gaussian likelihood with a non-conjugate Negative Binomial likelihood to capture the over-dispersion in COVID-19 death counts. All methods use $M \in \{15, 30\}$ inducing points and are trained for 5000 iterations per task with a learning rate of 0.01. Figure 4 reports the change of NLPD through online learning for the first four out of five tasks. The wide metric variance reflects the noisy nature of death-count data as it is difficult to accurately track down COVID-19 death counts. OHSVGP achieves the best performance overall while OSVGP forgets Task 1 with small $M$.

**Runtime comparison.** Table 1 shows the accumulated wall-clock runtime for different methods to learn all the tasks. Unlike OSVGP and OVC-optZ, which must iteratively optimize inducing points (for which we train e.g., 1000 iterations for time-series regression tasks), OHSVGP, OVFF (both based on interdomain inducing points), and OVC (based on one-time pivoted-Cholesky update of inducing points for each task) bypass this cumbersome optimization. In particular, OHSVGP recurrently evolves $\mathbf{K_{fu}}$ and $\mathbf{K_{uu}}$ for each new task. For regression problems where closed-form posterior can be obtained, OHSGPR requires no training at all. As a result, OHSGPR, OVC and OVFF run significantly faster, adapting to all tasks within a couple of seconds for Solar Irradiance and Audio data. For COVID data, even when free-form variational parameters of

Table 1: Wall-clock accumulated runtime for learning all the tasks on a single NVIDIA RTX3090 GPU in seconds, of different models for time series prediction experiments.

| Method | Solar Irradiance M=50 | Solar Irradiance M=150 | Audio Data M=100 | Audio Data M=200 | COVID M=15 | COVID M=30 |
|---|---|---|---|---|---|---|
| OSGPR/OSVGP | 140 | 149 | 144 | 199 | 525 | 530 |
| OVC | 0.450 | 0.620 | 0.558 | 0.863 | 345 | 360 |
| OVFF | 0.327 | 0.354 | 0.295 | 0.356 | - | - |
| OHSGPR/OHSVGP | 0.297 | 0.394 | 0.392 | 0.655 | 370 | 380 |

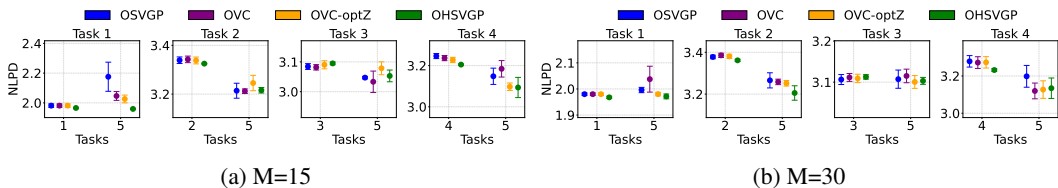

(a) M=15          (b) M=30

Figure 4: Test set NLPD on COVID dataset right after learning Task $i$ and after learning all the tasks.

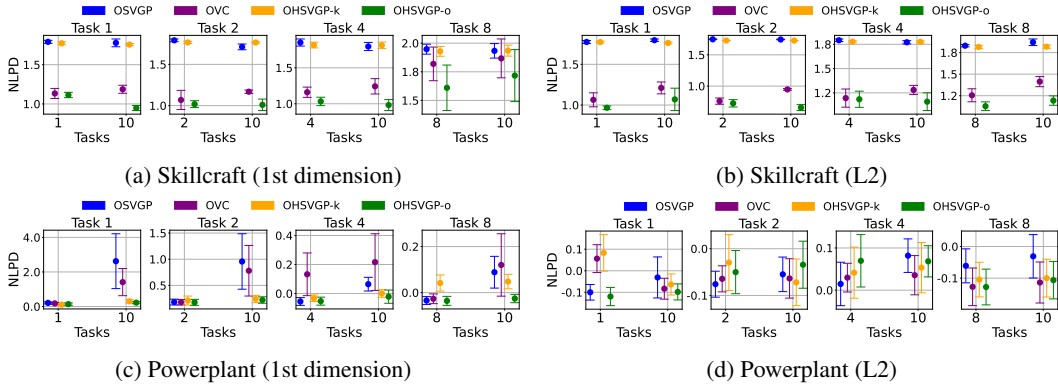

(a) Skillcraft (1st dimension)        (b) Skillcraft (L2)

(c) Powerplant (1st dimension)       (d) Powerplant (L2)

Figure 5: Test set NLPD after continually learning Task $i$ and after learning all the tasks for $i = 1$, 2, 4, 8. Tasks are created by splitting Powerplant and Skillcraft datasets with inputs sorted either according to the 1st input dimension or L2 distance to the origin).

inducing variables are learned using uncollapsed ELBO, OHSVGP and OVC are still significantly faster than OSVGP since no gradient computation is required for the inducing points.

## 5.2 Continual learning on UCI datasets

We use 256 inducing variables for all methods, and for each task, we train each method for 2000 iterations with a learning rate of 0.005. We only consider OVC here since initial trials show OVC-optZ give worse results on these two datasets. As described in Section 3.3, within each task, OHSVGP requires sorting the data points to compute prior covariance matrices via recurrence. We consider two sorting criteria. The first one, which we call OHSVGP-o, uses the oracle order compatible with how the tasks are created (e.g., sort with L2-distance to the origin if the tasks are initially splitted based on it). In real-world problems, we typically do not have the information on how the distribution shifts from task to task. Hence, we also consider OHSVGP-k, which uses a heuristic sorting method based on kernel similarity: we select the $i$-th point in task $j$ to be $\boldsymbol{x}_i^{(j)} = \arg\max_{\boldsymbol{x} \in \boldsymbol{X}^{(j)}} k(\boldsymbol{x}, \boldsymbol{x}_{i-1}^{(j)})$ for $i > 1$, and the first point in first task is set to be $\boldsymbol{x}_1^{(1)} = \arg\max_{\boldsymbol{x} \in \boldsymbol{X}^{(1)}} k(\boldsymbol{x}, \boldsymbol{0})$. Figure 5 compares the two variants of OHSVGP with OSVGP and OVC. Overall, OSVGP achieves the worst performance and is again prone to forgetting the older tasks, especially in Figure 5c. OVC performs decently for Skillcraft but it also demonstrates catastrophic forgetting in Figure 5c. While OHSVGP-k achieves similar performance as OSVGP on Skillcraft, OHSVGP-o consistently outperforms the other methods across all 4 scenarios, suggesting the importance of a sensible sorting method when applying OHSVGP for continual learning. Here, we only report the results for Task 1, 2, 4, and 8 for concise presentation, and in Appendix F.1, we include the complete results for all the tasks (the overall conclusion is the same). In Appendix F.3, we further visualize how different sorting methods impact OHSVGP's performance in continual learning with a 2D continual classification problem.

## 5.3 Continual learning for high dimensional time series prediction

All models share a two-layer MLP encoder–decoder, a 20-dimensional latent space, and a multi-output GP with independent components; we use $M \in \{50, 100\}$ and train each task for 20 epochs with learning rate 0.005 on single NVIDIA A6000 GPU. The continual learning in SVGPVAE is achieved

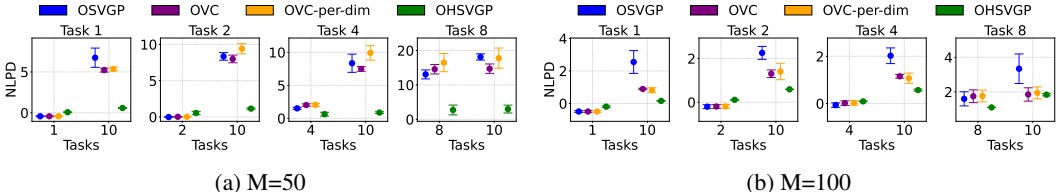

(a) M=50            (b) M=100

Figure 6: Test set NLPD after continually learning Task $i$ and after learning all the tasks for $i = 1, 2, 4, 8$, on ERA5 dataset.

by further imposing Elastic Weight Consolidation (EWC; [Kirkpatrick et al., 2017]) loss on the encoder and decoder, which yields the vanilla baseline, Online SVGPVAE (OSVGP). Since EWC alone leaves inducing locations non-regularized, a principled online placement rule for the inducing points will improve the model. Thus, we further consider OVC-SVGPVAE (OVC) which adjusts inducing points online via Pivoted-Cholesky, and OVC-SVGPVAE per dimension (OVC-per-dim), which makes OVC more flexible by allocating a separate set of $M$ inducing points to every latent dimension. Our method, Online HiPPO SVGPVAE (OHSVGP), replaces standard inducing points in SVGPVAE with HiPPO inducing variables and updates them online via recurrence. Figure 6 plots the change of NLPD during continual learning for Task 1, 2, 4, and 8 (full results in Appendix F.1). The performance of OHSVGP remains stable throughout, while the other methods all demonstrate obvious catastrophic forgetting shown by the large gaps between performances after learning current task $i$ and after learning final task 10. Two factors plausibly explain the gap: first, standard inducing points cannot adequately cover the long time axis, whereas OHSVGP ties its inducing variables to basis functions rather than time locations; second, the added encoder–decoder complexity makes optimization harder for models that must reuse a limited inducing set. Increasing $M$ narrows the gap but scales at $\mathcal{O}(M^3)$ computational and $\mathcal{O}(M^2)$ memory cost respectively, underscoring OHSVGP's superior efficiency.

## 6 Conclusion

We introduce OHSVGP, a novel online Gaussian process model that leverages the HiPPO framework for robust long-range memory in online/continual learning. By interpreting HiPPO's time-varying orthogonal projections as adaptive interdomain GP basis functions, we leverage SSM for improved online GP. This connection allows OHSVGP to harness HiPPO's efficient ODE-based recurrent updates while preserving GP-based uncertainty-aware prediction. Empirical results on a suite of online and continual learning tasks show that OHSVGP outperforms existing online sparse GP methods, especially in scenarios requiring long-term memory. Moreover, its recurrence-based covariance updates yield far lower computational overhead than OSVGP's sequential inducing point optimization. This efficient streaming capability and preservation of historical information make OHSVGP well-suited for real-world applications demanding both speed and accuracy.

**Broader impact.**  This paper presents work whose goal is to advance machine learning research. There may exist potential societal consequences of our work, however, none of which we feel must be specifically highlighted here.

## Acknowledgments and Disclosure of Funding

Samir Bhatt acknowledges funding from the MRC Centre for Global Infectious Disease Analysis (reference MR/X020258/1), funded by the UK Medical Research Council (MRC). This UK funded award is carried out in the frame of the Global Health EDCTP3 Joint Undertaking. Samir Bhatt acknowledges support from the Danish National Research Foundation via a chair grant (DNRF160) which also supports Jacob Curran-Sebastian. Samir Bhatt acknowledges support from The Eric and Wendy Schmidt Fund For Strategic Innovation via the Schmidt Polymath Award (G-22-63345) which also supports Harrison Bo Hua Zhu. Samir Bhatt acknowledges support from the Novo Nordisk Foundation via The Novo Nordisk Young Investigator Award (NNF20OC0059309).

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

# A HiPPO-LegS matrices

Here we provide the explicit form of matrices used in our implementation of HiPPO-LegS [Gu et al., 2020]. For a given time $t$, the measure $\omega^{(t)}(x) = \frac{1}{t}\mathbf{1}_{[0,t]}(x)$ and basis functions $\phi_m^{(t)}(x) = g_m^{(t)}(x)\omega^{(t)}(x) = \frac{\sqrt{2m+1}}{t}P_m\left(\frac{2x}{t} - 1\right)\mathbf{1}_{[0,t]}(x)$ are used, where $P_m(\cdot)$ is the $m$-th Legendre polynomial and $\mathbf{1}_{[0,t]}(x)$ is the indicator function on the interval $[0,t]$. These basis functions are orthonormal, i.e.,

$$\int_0^t \frac{g_m^{(t)}(x)g_n^{(t)}(x)}{t}\mathrm{d}x = \delta_{mn} \tag{6}$$

Following Gu et al. [2020], the HiPPO-LegS framework maintains a coefficient vector $\mathbf{c}(t) \in \mathbb{R}^{M \times 1}$ that evolves according to the ODE:

$$\frac{\mathrm{d}}{\mathrm{d}t}\mathbf{c}(t) = \mathbf{A}(t)\mathbf{c}(t) + \mathbf{B}(t)f(t) \tag{7}$$

where $f(t)$ is the input signal at time $t$. The matrices $\mathbf{A}(t) \in \mathbb{R}^{M \times M}$ and $\mathbf{B}(t) \in \mathbb{R}^{M \times 1}$ are given by:

$$[\mathbf{A}(t)]_{nk} = \frac{1}{t}\mathbf{A}, \quad [\mathbf{A}]_{nk} = \begin{cases} -\sqrt{(2n+1)(2k+1)} & \text{if } n > k \\ -n+1 & \text{if } n = k \\ 0 & \text{if } n < k \end{cases} \tag{8}$$

and

$$[\mathbf{B}(t)]_n = \frac{[\mathbf{B}]_n}{t} = \frac{\sqrt{2n+1}}{t} \tag{9}$$

These matrices govern the evolution of the basis function coefficients over time, where the factor $1/t$ reflects the time-dependent scaling of the basis functions to the adaptive interval $[0,t]$. When discretized, this ODE yields the recurrence update used in our implementation.

# B Computing prior covariance of the inducing variables $\mathbf{K}_{\mathbf{uu}}^{(t)}$

We provide the detailed derivation for the following two approaches when the inducing functions are defined via HiPPO-LegS. Recall that the $\ell m$-th element of the prior covariance matrix for the inducing variables is given by

$$[\mathbf{K}_{\mathbf{uu}}^{(t)}]_{\ell m} = \iint k(x,x')\phi_\ell^{(t)}(x)\phi_m^{(t)}(x')\mathrm{d}x\mathrm{d}x', \tag{10}$$

where $\phi_\ell^{(t)}(x) = g_\ell^{(t)}(x)\omega^{(t)}(x)$ are the time-varying basis functions under the HiPPO-LegS framework.

## B.1 RFF approximation

Since $k(x,x')$ depends on both $x$ and $x'$, a recurrence update rule based on the original HiPPO formulation, which is designed for single integral, can not be obtained directly for $\boldsymbol{K}_{uu}^{(t)}$. Fortunately, for stationary kernels, Bochner Theorem [Rudin, 1994] can be applied to factorize the above double integrals into two separate single integrals, which gives rise to Random Fourier Features (RFF) approximation [Rahimi and Recht, 2007]: for a stationary kernel $k(x,x') = k(|x - x'|)$, RFF approximates it as follows:

$$k(x,x') = \mathbb{E}_{p(w)}\Big[\cos(wx)\cos(wx') + \sin(wx)\sin(wx')\Big]$$
$$\approx \frac{1}{N}\sum_{n=1}^N \left[\cos\left(w_n x\right)\cos\left(w_n x'\right) + \sin\left(w_n x\right)\sin\left(w_n x'\right)\right], \tag{11}$$

where $w_n \sim p(w)$ is the spectral density of the kernel. Substituting this into the double integral (Eq. 10) factorizes the dependency on $x$ and $x'$, reducing $[\mathbf{K}_{\mathbf{uu}}^{(t)}]_{\ell m}$ to addition of products of one-dimensional integrals. Each integral based on a Monte Carlo sample $w$ has the form of either

$$Z_{w,\ell}^{(t)} = \int \cos(wx)\phi_\ell^{(t)}(x)\,\mathrm{d}x \quad \text{or} \quad Z_{w,\ell}'^{(t)} = \int \sin(wx)\phi_\ell^{(t)}(x)\,\mathrm{d}x, \tag{12}$$

which corresponds to a standard projection coefficient in the HiPPO framework. We further stack these integrals based on $M$ basis functions and define

$$\mathbf{Z}_w^{(t)} = \left[ Z_{w,1}^{(t)}, \cdots, Z_{w,M}^{(t)} \right]^\top, \quad \mathbf{Z}_w'^{(t)} = \left[ Z_{w,1}'^{(t)}, \cdots, Z_{w,M}'^{(t)} \right]^\top. \tag{13}$$

Collecting $N$ Monte Carlo samples $\{w_n\}_{n=1}^N$, we form the feature matrix

$$\mathbf{Z}^{(t)} = \begin{bmatrix} \mathbf{Z}_{w_1}^{(t)} & \mathbf{Z}_{w_2}^{(t)} & \cdots & \mathbf{Z}_{w_N}^{(t)} & \mathbf{Z}_{w_1}'^{(t)} & \mathbf{Z}_{w_2}'^{(t)} & \cdots & \mathbf{Z}_{w_N}'^{(t)} \end{bmatrix}, \tag{14}$$

and the RFF approximation of the covariance is

$$\mathbf{K}_{\mathbf{uu}}^{(t)} \approx \frac{1}{N} \mathbf{Z}^{(t)} \left( \mathbf{Z}^{(t)} \right)^\top. \tag{15}$$

Since $\mathbf{Z}_{w_n}^{(t)}$ and $\mathbf{Z}_{w_n}'^{(t)}$ are standard HiPPO projection coefficient, their computation is governed by the HiPPO ODE evolution as before

$$\frac{\mathrm{d}}{\mathrm{d}t}\mathbf{Z}_{w_n}^{(t)} = \mathbf{A}(t)\,\mathbf{Z}_{w_n}^{(t)} + \mathbf{B}(t)\,h_n(t), \quad \frac{\mathrm{d}}{\mathrm{d}t}\mathbf{Z}_{w_n}'^{(t)} = \mathbf{A}(t)\,\mathbf{Z}_{w_n}'^{(t)} + \mathbf{B}(t)\,h_n'(t), \tag{16}$$

with $h_n(t) = \cos(w_n t)$ and $h_n'(t) = \sin(w_n t)$ and these ODEs can be solved in parallel across different Monte Carlo samples. In summary, the procedure involves sampling multiple random features, updating them recurrently to time $t$, and averaging across samples to obtain RFF approximation of $\mathbf{K}_{\mathbf{uu}}^{(t)}$.

For non-stationary kernels, more advanced Fourier feature approximation techniques (e.g., [Ton et al., 2018]) can be applied.

## B.2 Direct ODE evolution

Differentiating $[\mathbf{K}_{\mathbf{uu}}^{(t)}]_{\ell,m}$ with respect to $t$ gives

$$\frac{d}{dt}[\mathbf{K}_{\mathbf{uu}}^{(t)}]_{\ell,m} = \iint k(x,x')\frac{\partial}{\partial t}\left[\phi_\ell^{(t)}(x)\phi_m^{(t)}(x')\right]\mathrm{d}x\mathrm{d}x'. \tag{17}$$

Applying the product rule:

$$\frac{\mathrm{d}}{\mathrm{d}t}[\mathbf{K}_{\mathbf{uu}}^{(t)}]_{\ell,m} = \iint k(x,x')\frac{\partial}{\partial t}\phi_\ell^{(t)}(x)\phi_m^{(t)}(x')\mathrm{d}x\mathrm{d}x' + \iint k(x,x')\phi_\ell^{(t)}(x)\frac{\partial}{\partial t}\phi_m^{(t)}(x')\mathrm{d}x\mathrm{d}x'. \tag{18}$$

In HiPPO-LegS, each $\phi_\ell^{(t)}(x)$ obeys an ODE governed by lower-order scaled Legendre polynomials on $[0,t]$ and a Dirac delta boundary term at $x = t$. Concretely,

$$\begin{aligned} \frac{\partial}{\partial t}\phi_\ell^{(t)}(x) = -\frac{\sqrt{2\ell+1}}{t}\Big[ & \frac{\ell+1}{\sqrt{2\ell+1}}\phi_\ell^{(t)}(x) + \sqrt{2\ell-1}\phi_{\ell-1}^{(t)}(x) \\ & + \sqrt{2\ell-3}\phi_{\ell-2}^{(t)}(x)\cdots\Big] + \frac{1}{t}\delta_t(x), \end{aligned} \tag{19}$$

where $\delta_t(x)$ is the Dirac delta at $x = t$ (see Appendix D.3 in Gu et al. [2020] for details).

Substituting this expression into the integrals yields the boundary terms of the form $\int k(t,x')\,\phi_m^{(t)}(x')\,\mathrm{d}x'$, along with lower-order terms involving $\{[\mathbf{K}_{\mathbf{uu}}^{(t)}]_{\ell,m}, [\mathbf{K}_{\mathbf{uu}}^{(t)}]_{\ell-1,m}, \cdots\}$, etc. Summarizing in matrix form leads to

$$\frac{\mathrm{d}}{\mathrm{d}t}\mathbf{K}_{\mathbf{uu}}^{(t)} = \left[ \mathbf{A}(t)\,\mathbf{K}_{\mathbf{uu}}^{(t)} + \mathbf{K}_{\mathbf{uu}}^{(t)}\mathbf{A}(t)^\intercal \right] + \frac{1}{t}\left[ \tilde{\mathbf{B}}(t) + \tilde{\mathbf{B}}(t)^\intercal \right], \tag{20}$$

where the $lm$-th entry of $\mathbf{K}_{\mathbf{uu}}^{(t)} \in \mathbb{R}^{M \times M}$ is $[\mathbf{K}_{\mathbf{uu}}^{(t)}]_{\ell,m}$, $\mathbf{A}(t) \in \mathbb{R}^{M \times M}$ is the same lower-triangular matrix from the HiPPO-LegS framework defined in Eq. 8, and $\tilde{\mathbf{B}}(t) \in \mathbb{R}^{M \times M}$ is built from the boundary contributions as

$$\tilde{\mathbf{B}}(t) = \mathbf{c}(t)\mathbf{1}_M, \tag{21}$$

where $\mathbf{1}_M \in \mathbb{R}^{1 \times M}$ is a row vector of ones of size $M$ and $\mathbf{c}(t) \in \mathbb{R}^{M \times 1}$ is the coefficient vector with each element being

$$c_\ell(t) = \int k(t, x) \, \phi_\ell^{(t)}(x) \, \mathrm{d}x. \tag{22}$$

After discretizing in $t$ (e.g. an Euler scheme), one repeatedly updates $\mathbf{K}_{\mathbf{uu}}^{(t)}$ and the boundary vector $\mathbf{c}(t)$ over time.

### B.2.1 Efficient computation of $\tilde{\mathbf{B}}(t)$

Computing $\tilde{\mathbf{B}}(t)$ directly at each time step requires evaluating $M$ integrals, which can be computationally intensive, especially when $t$ changes incrementally and we need to update the matrix $\tilde{\mathbf{B}}(t)$ repeatedly.

To overcome this inefficiency, we propose an approach that leverages the HiPPO framework to compute $\tilde{\mathbf{B}}(t)$ recursively as $s$ evolves. This method utilizes the properties of stationary kernels and the structure of the Legendre polynomials to enable efficient updates.

**Leveraging Stationary Kernels** Assuming that the kernel $k(x, t)$ is stationary, it depends only on the difference $d = |x - t|$, so $k(x, t) = k(d)$. In our context, since we integrate over $x \in [t_{\text{start}}, t]$ with $x \leq t$, we have $d = t - x \geq 0$. Therefore, we can express $k(x, t)$ as a function of $d$ over the interval $[0, t - t_{\text{start}}]$:

$$k(x, t) = k(t - x) = k(d), \quad \text{with} \quad d \in [0, t - t_{\text{start}}]. \tag{23}$$

Our goal is to approximate $k(d)$ over the interval $[0, t - t_{\text{start}}]$ using the orthonormal Legendre basis functions scaled to this interval. Specifically, we can represent $k(d)$ as

$$k(d) \approx \sum_{m=0}^{M-1} \tilde{c}_m(t) \, g_m^{(t)}(d), \tag{24}$$

where $g_m^{(t)}(d)$ are the Legendre polynomials rescaled to the interval $[0, t - t_{\text{start}}]$.

**Recursive Computation via HiPPO-LegS** To efficiently compute the coefficients $\tilde{c}_m(t)$, we utilize the HiPPO-LegS framework, which provides a method for recursively updating the coefficients of a function projected onto an orthogonal basis as the interval expands. In our case, as $t$ increases, the interval $[t_{\text{start}}, t]$ over which $k(d)$ is defined also expands, and we can update $\tilde{c}_m(t)$ recursively.

Discretizing time with step size $\Delta t$ and indexing $t_k = t_{\text{start}} + k\Delta t$, the update rule using the Euler method is:

$$\tilde{\mathbf{c}}_{k+1} = \left(\mathbf{I} - \frac{1}{k}\mathbf{A}\right)\tilde{\mathbf{c}}_k + \frac{1}{k}\mathbf{B}\,k(t_k), \tag{25}$$

where $\tilde{\mathbf{c}}_k = [\tilde{c}_0(t_k), \tilde{c}_1(t_k), \ldots, \tilde{c}_{M-1}(t_k)]^\intercal$, and $\mathbf{A} \in \mathbb{R}^{M \times M}$ and $\mathbf{B} \in \mathbb{R}^M$ are again matrices defined by the HiPPO-LegS operator as in Eq. 8 and 9.

**Accounting for Variable Transformation and Parity** The change of variables from $x$ to $d = t - x$ introduces a reflection in the function domain. Since the Legendre polynomials have definite parity, specifically,

$$P_m(-x) = (-1)^m P_m(x), \tag{26}$$

we need to adjust the coefficients accordingly when considering the reflected function.

As a result of this reflection, when projecting $k(d)$ onto the Legendre basis, the coefficients $\tilde{c}_m(t)$ computed via the HiPPO-LegS updates will correspond to a reflected version of the function. To

account for this, we apply a parity correction to the coefficients. Specifically, the corrected coefficients $c_m(t)$ are related to $\tilde{c}_m(t)$ by a sign change determined by the degree $m$:

$$c_m(t) = (-1)^m \tilde{c}_m(t). \tag{27}$$

This parity correction ensures that the computed coefficients properly represent the function over the interval $[t_{\text{start}}, t]$ without the effect of the reflection.

By computing $\mathbf{c}(t)$ recursively as $t$ evolves, we can efficiently update $\tilde{\mathbf{B}}(t) = \mathbf{c}(t)[1, \ldots, 1]$ at each time step without the need to evaluate the integrals directly. This approach significantly reduces the computational burden associated with updating $\tilde{\mathbf{B}}(t)$ and allows for efficient computation of $\mathbf{K}_{\mathbf{uu}}^{(t)}$ via the ODE.

### B.2.2   Unstability of directly evolving $\mathbf{K}_{\mathbf{uu}}^{(t)}$ as ODE.

Empirically, we find that the direct ODE approach is less stable compared with RFF approach. Intuitively, it can be seen from the difference in the forms of their evolutions, especially in the first term. In RFF approach, the first term of the evolution of Fourier feature is of the form $\mathbf{A}(t)\,\mathbf{Z}_w^{(t)}$, which includes evolving vectors with the operator $\mathcal{L}_1 : \mathbf{X} \to \mathbf{A}(t)\mathbf{X}$. In direct ODE approach, the first term in the direct evolution of $\mathbf{K}_{\mathbf{uu}}^{(t)}$ is of the form $\mathbf{A}(t)\,\mathbf{K}_{\mathbf{uu}}^{(t)} + \mathbf{K}_{\mathbf{uu}}^{(t)}\mathbf{A}(t)^\top$, which requires the Lyapunov operator $\mathcal{L}_2 : \mathbf{X} \to \mathbf{A}(t)\mathbf{X} + \mathbf{X}\mathbf{A}(t)^\top$. The critical difference is that $\mathcal{L}_2$ has eigenvalues $\{\lambda_i + \lambda_j\}$ (where $\lambda_i$ and $\lambda_j$ are eigenvalues of $\mathbf{A}(t)$) [Horn and Johnson, 1991], while $\mathcal{L}_1$ has eigenvalues $\{\lambda_i\}$. Since HiPPO-LegS uses a lower-triangular $\mathbf{A}(t)$ with negative diagonal entries, the eigenvalues are all negative $\lambda_i < 0$. Hence, the eigenvalues of the Lyapunov operator $\mathcal{L}_2$ are approximately as twice negative as the eigenvalues of $\mathcal{L}_1$, leading to a stiff ODE system with poorer numerical conditioning.

## C   Finite basis approximation of posterior OHSVGP

Here, we show that $q(\mathbf{u}^{(t)})$ is the distribution of HiPPO coefficients of the posterior OHSVGP $q_t(f)$. From Eq. 2, the posterior of the function values evaluated at arbitrary indices $\mathbf{X}$ is $q_t(\mathbf{f_X}) = \mathcal{N}(\mathbf{f_X}; \mathbf{K}_{\mathbf{f_X u}}^{(t)}\mathbf{K}_{\mathbf{uu}}^{(t)-1}\mathbf{m}_{\mathbf{u}}^{(t)}, \mathbf{K}_{\mathbf{f_X f_X}} - \mathbf{K}_{\mathbf{f_X u}}^{(t)}\mathbf{K}_{\mathbf{uu}}^{(t)-1}[\mathbf{K}_{\mathbf{uu}}^{(t)} - \mathbf{S}_{\mathbf{u}}^{(t)}]\mathbf{K}_{\mathbf{uu}}^{(t)-1}\mathbf{K}_{\mathbf{uf_X}}^{(t)})$.

Based on this, we compute the mean of the $m$-th HiPPO coefficient for $q_t(f)$ as follows,

$$
\begin{aligned}
&E_{q_t(f)}\left[\int f(x)\phi_m^{(t)}(x)dx\right] \\
&= \int E_{q_t(f)}\left[f(x)\right]\phi_m^{(t)}(x)dx \\
&= \left(\int \mathbf{K}_{f_x\mathbf{u}}^{(t)}\mathbf{K}_{\mathbf{uu}}^{(t)-1}\phi_m^{(t)}(x)dx\right)\mathbf{m}_{\mathbf{u}}^{(t)} \\
&= \left[\int\int k(x,x')\begin{pmatrix}\phi_1^{(t)}(x')\\ \cdots \\ \phi_M^{(t)}(x')\end{pmatrix}dx'\mathbf{K}_{\mathbf{uu}}^{(t)-1}\phi_m^{(t)}(x)dx\right]\mathbf{m}_{\mathbf{u}}^{(t)} \\
&= \left[\int\int k(x,x')\begin{pmatrix}\phi_1^{(t)}(x')\phi_m^{(t)}(x)\\ \cdots \\ \phi_M^{(t)}(x')\phi_m^{(t)}(x)\end{pmatrix}dx'dx\right]\mathbf{K}_{\mathbf{uu}}^{(t)-1}\mathbf{m}_{\mathbf{u}}^{(t)} \\
&= \left[\mathbf{K}_{\mathbf{uu}}^{(t)}\right]_{m,:}\mathbf{K}_{\mathbf{uu}}^{(t)-1}\mathbf{m}_{\mathbf{u}}^{(t)} \\
&= \left[\mathbf{m}_{\mathbf{u}}^{(t)}\right]_m,
\end{aligned}
\tag{28}
$$

which is exactly the variational mean of $q(u_m^{(t)})$. Similarly, the covariance between the $l$-th and the $m$-th HiPPO coefficient for $q_t(f)$ can be computed as

$$
E_{q_t(f)}\left[\left(\int f(x)\phi_l^{(t)}(x)dx\right)\left(\int f(x')\phi_m^{(t)}(x')dx'\right)\right] - \left[\mathbf{m_u}^{(t)}\right]_l\left[\mathbf{m_u}^{(t)}\right]_m
$$
$$
= \int\int E_{q_t(f)}\left[f(x)f(x')\right]\phi_l^{(t)}(x)\phi_m^{(t)}(x')dxdx' - \left[\mathbf{m_u}^{(t)}\right]_l\left[\mathbf{m_u}^{(t)}\right]_m
$$
$$
= \int\int\left(k(x,x') - \mathbf{K}_{f_x\mathbf{u}}^{(t)}\mathbf{K_{uu}}^{(t)-1}[\mathbf{K_{uu}}^{(t)} - \mathbf{S_u}^{(t)}]\mathbf{K_{uu}}^{(t)-1}\mathbf{K}_{\mathbf{u}f_{x'}}^{(t)}\right)\phi_l^{(t)}(x)\phi_m^{(t)}(x')dxdx'
$$
$$
\quad + \int\int E_{q_t(f)}\left[f(x)\right]E_{q_t(f)}\left[f(x')\right]\phi_l^{(t)}(x)\phi_m^{(t)}(x')dxdx' - \left[\mathbf{m_u}^{(t)}\right]_l\left[\mathbf{m_u}^{(t)}\right]_m
$$
$$
= \left[\mathbf{K_{uu}}^{(t)}\right]_{lm} - \left(\int\mathbf{K}_{f_x\mathbf{u}}^{(t)}\mathbf{K_{uu}}^{(t)-1}\phi_l^{(t)}(x)dx\right)[\mathbf{K_{uu}}^{(t)} - \mathbf{S_u}^{(t)}]\left(\int\phi_m^{(t)}(x')\mathbf{K_{uu}}^{(t)-1}\mathbf{K}_{\mathbf{u}f_{x'}}^{(t)}dx'\right)
$$
$$
= \left[\mathbf{K_{uu}}^{(t)}\right]_{lm} - \left[\mathbf{K_{uu}}^{(t)}\right]_{l,:}\mathbf{K_{uu}}^{(t)-1}\mathbf{K_{uu}}^{(t)}\mathbf{K_{uu}}^{(t)-1}\left[\mathbf{K_{uu}}^{(t)}\right]_{:,m} + \left[\mathbf{K_{uu}}^{(t)}\right]_{l,:}\mathbf{K_{uu}}^{(t)-1}\mathbf{S_u}^{(t)}\mathbf{K_{uu}}^{(t)-1}\left[\mathbf{K_{uu}}^{(t)}\right]_{:,m}
$$
$$
= \left[\mathbf{S_u}^{(t)}\right]_{lm},
$$

$$(29)$$

which is exactly the variational covariance between $u_l^{(t)}$ and $u_m^{(t)}$ in $q(\mathbf{u}^{(t)})$.

Hence, if $f \sim q_t(f)$, then $q(u_m^{(t)}) \overset{d}{=} \int f(x)\phi_m^{(t)}(x)\mathrm{d}x$, which implies that we can approximate the posterior OHSVGP with finite basis: $f = \sum_{m=1}^M u_m^{(t)}\phi_m^{(t)}(x)$, $u_m^{(t)} \sim q(u_m^{(t)})$.

## D  Additional experimental details

### D.1  Infectious disease modeling

For a sanity check, we also fit a weekly renewal-equation model [Flaxman et al., 2020, Unwin et al., 2020, Monod et al., 2021], trained offline on the full history. This is a well-established infectious disease model that has been widely utilized by scientists during the COVID-19 pandemic, and we include its results in section F.1 (denoted as AR(2) Renewal in the legend). Interestingly, OHSVGP achieves better predictive performance than this traditional infectious disease model. Although this may be partly due to the strong inductive biases of the renewal equations, it nevertheless highlights OHSVGP's suitability for long infectious-disease time series.

The details of renewal-equation model are as follows. Let $y_{t,a}$ be the number of deaths on day $t = 1, \ldots, T$ in state $a = 1, \ldots, A$ (in our experiments, $a = 1$ since we only fit on 1 state), The probabilistic model is:

$$
(\text{expected deaths}), \quad \mu_{t,a} = \sum_{\tau=1}^{t-1} m_{\tau,a}\text{IFR}(t - \tau) \tag{30}
$$

$$
(\text{expected infections}), \quad m_{\tau,a} = R_{\tau,a}^{\text{adj}}\sum_{i=1}^{\tau-1} m_{i,a}\text{SI}(\tau - i) \tag{31}
$$

$$
(\text{Adjusted reproductive number}), \quad R_{\tau,a}^{\text{adj}} = \frac{N_a - C_{\tau,a}}{N_a}R_{\tau,a} \tag{32}
$$

$$
R_{t,a} = 3.3 \times 2\sigma(-f_a(\tau)) \tag{33}
$$

$$
(\text{cumulative infections}), \quad C_{\tau,a} = \text{NumDeathsInit}_a + \sum_{i=1}^{\tau-1} C_{i,a}, \tag{34}
$$

where $\sigma$ is the sigmoid function, $N_a$ is the population of country $a$, $\text{NumDeathsInit}_a$ is the initial number of deaths in country $a$, the IFR is the infection fatality rate and SI is the serial interval. The choice of $f_a(\tau)$ requires a stochastic process and here we choose to model it as weekly random

effect on day $\tau$ using AR(2) process as in Monod et al. [2021], Unwin et al. [2020]. Typically, Markov chain Monte Carlo (MCMC) methods are used to infer the posterior distribution of $f_a$ and the other parameters, so it is less scalable than the online sparse GP based models shown in main text, especially when the number of states $a$ and time steps $T$ increase.

## D.2 SVGPVAE model details

With SVGPVAE, we utilize Jazbec et al. [2021] and notation from Zhu et al. [2023], and we have the following encoder-decoder model:

$$p(\mathbf{y}_{1:T}) = p(\mathbf{f}_{1:T}) \prod_{t=1}^{T} p(\mathbf{y}_t | \mathbf{f}_t), \tag{35}$$

with likelihood $p(\mathbf{y}_t | \mathbf{f}_t) = \mathcal{N}(\mathbf{y}_t | \varphi(\mathbf{f}_t), \sigma^2 I)$ and decoder network $\varphi : \mathbb{R}^L \to \mathbb{R}^{d_y}$. The encoder $\phi : \mathbb{R}^{d_y} \to \mathbb{R}^{2L}$ yields $(\tilde{\mathbf{y}}_t^{1:L}, \tilde{\mathbf{v}}_t^{1:L}) = \phi(\mathbf{y}_t)$. $\mathbf{f}_t$ follows an $L$-dimensional multi-output GP and its approximate posterior is given by:

$$q(\mathbf{f}_{1:T}) = \prod_{l=1}^{L} p(\mathbf{f}_{1:T}^l | \mathbf{u}_m^l) q(\mathbf{u}^l), \quad q(\mathbf{u}^l) = \mathcal{N}(\mathbf{u}^l | \mathbf{m}^l, \mathbf{A}^l), \tag{36}$$

$$\mathbf{S}^l = \mathbf{K}_{\mathbf{uu}}^l + \mathbf{K}_{\mathbf{uf}}^l \mathrm{diag}(\tilde{\mathbf{v}}_{1:T}^l)^{-1} \mathbf{K}_{\mathbf{fu}}^l, \quad \mathbf{m}^l = \mathbf{K}_{\mathbf{uu}}^l (\mathbf{S}^l)^{-1} \mathbf{K}_{\mathbf{uf}}^l \mathrm{diag}(\tilde{\mathbf{v}}_{1:T}^l)^{-1} \tilde{\mathbf{y}}_{1:T}^l, \tag{37}$$

$$\mathbf{A}^l = \mathbf{K}_{\mathbf{uu}}^l (\mathbf{S}^l)^{-1} \mathbf{K}_{\mathbf{uu}}^l, \tag{38}$$

where $p(\mathbf{f}_{1:T}^l | \mathbf{u}_m^l)$ is the prior conditional distribution.

Following Jazbec et al. [2021], the objective function is defined as:

$$\mathcal{L}_{\mathrm{SVGPVAE}}(\theta) = \sum_{t=1}^{T} \left[ \mathbb{E}_{q(\mathbf{f}_t)} \log p(\mathbf{y}_t | \mathbf{f}_t) - \log \mathcal{N}(\mathbf{f}_t | \tilde{\mathbf{y}}_t, \tilde{\mathbf{v}}_t) \right] + \sum_{l=1}^{L} \mathcal{L}_H^l, \tag{39}$$

where $\mathcal{L}_H^l$ is the "Hensman" ELBO described in Equation 7 of Jazbec et al. [2021]. Since the variational parameters $\mathbf{m}^l$ and $\mathbf{S}^l$, and the likelihood are all amortized by neural networks, we further add EWC (Elastic Weight Consolidation; [Kirkpatrick et al., 2017]) regularization for both encoder and decoder networks to the loss above for continual learning.

# E  Algorithmic Breakdown of OHSVGP

---

**Algorithm 1** The HIPPO-SVGP ELBO for a single task of data. Differences with SVGP in blue.

---

**Require:**   • $\mathbf{X} = \{x_1, \ldots x_n = t_1\}$ (training time steps up to time $t_1$),
   • $\{y_1, \ldots, y_n\}$ (training targets),
   • $\mathbf{Z} \in \mathbb{R}^{M \times 1}$ (inducing points) $A(t) \in \mathbb{R}^{M \times M}, B(t) \in \mathbb{R}^{M \times 1}$ (HIPPO matrices) ,
   • $\mathbf{m}_{\mathbf{u}} \in \mathbb{R}^{M \times 1}, \mathbf{S}_{\mathbf{u}} \in \mathbb{R}^{M \times M}$ (variational params)
1: $\mathbf{K}_{\mathbf{fu}} = k(\mathbf{X}, \mathbf{Z}), \mathbf{K}_{\mathbf{uu}} = k(\mathbf{Z}, \mathbf{Z})$, $\mathbf{K}_{\mathbf{fu}}^{t_1}, \mathbf{K}_{\mathbf{uu}}^{t_1}$ from HIPPO ODEs evolved from 0 to the final time step $t_1$ with HIPPO matrices $A(t), B(t)$
2: $\mu(x_i) = \mathbf{K}_{f_i \mathbf{u}}^{t_1} (\mathbf{K}_{\mathbf{uu}}^{t_1})^{-1} \mathbf{m}_{\mathbf{u}}$                                ▷ Variational Posterior Mean
3: $\sigma^2(x_i) = \mathbf{K}_{f_i f_i}^{t_1} - \mathbf{K}_{f_i \mathbf{u}}^{t_1} (\mathbf{K}_{\mathbf{uu}}^{t_1})^{-1} [\mathbf{K}_{\mathbf{uu}}^{t_1} - \mathbf{S}_{\mathbf{u}}] (\mathbf{K}_{\mathbf{uu}}^{t_1})^{-1} \mathbf{K}_{\mathbf{u} f_i}^{t_1}$   ▷ Variational Posterior Variance
4: $\ell_{\mathrm{varexp}} \leftarrow \sum_{i=1}^{n} \mathbb{E}_{\mathcal{N}(\mu(x_i), \sigma^2(x_i))} \left[ \log p(y_i \mid f_i) \right]$                ▷ closed form or quadrature/MC
5: $\mathrm{KL} \leftarrow \mathrm{KL}(\mathcal{N}(\mathbf{m}_{\mathbf{u}}, \mathbf{S}_{\mathbf{u}}) || \mathcal{N}(0, \mathbf{K}_{\mathbf{uu}}^{t_1}))$
6: **return** $\ell_{\mathrm{varexp}} - \mathrm{KL}$

---

**Algorithm 2** The OHSVGP ELBO on the second task after learning the first task. Differences with OSVGP in blue.

**Require:**
- $\mathbf{X}' = \{t_1 < x'_1, \ldots x'_{n'} = t_2\}$ (training time steps up to time $t_2$),
- $\{y'_1, \ldots, y'_{n'}\}$ (training targets),
- $\mathbf{Z}_{t_1} \in \mathbb{R}^{M \times 1}$ (frozen and learned inducing points from task 1) with inducing variables $\mathbf{u}_{t_1} = f(\mathbf{Z}_{t_1})$,
- $\mathbf{Z}_{t_2} \in \mathbb{R}^{M \times 1}$ (new inducing points for task 2) with inducing variables $\mathbf{u}_{t_2} = f(\mathbf{Z}_{t_2})$, $A(t) \in \mathbb{R}^{M \times M}, B(t) \in \mathbb{R}^{M \times 1}$ (HIPPO matrices),
- $\mathbf{m}_{\mathbf{u}_{t_1}} \in \mathbb{R}^{M \times 1}, \mathbf{S}_{\mathbf{u}_{t_1}} \in \mathbb{R}^{M \times M}$ (learned variational params from task 1),
- $\mathbf{m}_{\mathbf{u}_{t_2}} \in \mathbb{R}^{M \times 1}, \mathbf{S}_{\mathbf{u}_{t_2}} \in \mathbb{R}^{M \times M}$ (new variational params for task 2),
- $\mathbf{K}^{t_1}_{\mathbf{u}_{t_1}\mathbf{u}_{t_1}}$

1: $\mathbf{K}_{\mathbf{f}'\mathbf{u}_{t_2}} = k(\mathbf{X}', \mathbf{Z}_{t_2})$, $\mathbf{K}_{\mathbf{u}_{t_2}\mathbf{u}_{t_2}} = k(\mathbf{Z}_{t_2}, \mathbf{Z}_{t_2})$, $\mathbf{K}^{t_2}_{\mathbf{f}'\mathbf{u}_{t_2}}$ evolved from $0$ to the final time step $t_2$, $\mathbf{K}^{t_2}_{\mathbf{u}_{t_2}\mathbf{u}_{t_2}}$ evolved from $t_1$ to the final time step $t_2$ with HIPPO matrices $A(t), B(t)$

2: $\mu_{t_2}(x'_i) = \mathbf{K}^{t_2}_{f'_i\mathbf{u}_{t_2}}(\mathbf{K}^{t_2}_{\mathbf{u}_{t_2}\mathbf{u}_{t_2}})^{-1}\mathbf{m}_{\mathbf{u}_{t_2}}$ ▷ Variational Posterior Mean

3: $\sigma^2_{t_2}(x'_i) = \mathbf{K}^{t_2}_{f'_i f'_i} - \mathbf{K}^{t_2}_{f'_i\mathbf{u}_{t_2}}(\mathbf{K}^{t_2}_{\mathbf{u}_{t_2}\mathbf{u}_{t_2}})^{-1}[\mathbf{K}^{t_2}_{\mathbf{u}_{t_2}\mathbf{u}_{t_2}} - \mathbf{S}_{\mathbf{u}_{t_2}}](\mathbf{K}^{t_2}_{\mathbf{u}_{t_2}\mathbf{u}_{t_2}})^{-1}\mathbf{K}^{t_1}_{\mathbf{u}_{t_2}f'_i}$ ▷ Variational Posterior Variance

4: $\ell_{\text{varexp}} \leftarrow \sum_{i=1}^{n'} \mathbb{E}_{\mathcal{N}(\mu_{t_2}(x_i),\sigma^2_{t_2}(x_i))}\big[\log p(y'_i \mid f'_i)\big]$ ▷ closed form or quadrature/MC

5: $\tilde{\mathbf{m}}_{t_1 t_2} = \mathbf{K}^{t_2}_{\mathbf{u}_{t_1}\mathbf{u}_{t_2}}\mathbf{K}^{t_2}_{\mathbf{u}_{t_2}\mathbf{u}_{t_2}}\mathbf{m}_{\mathbf{u}_{t_2}}$ ▷ Mean of $\tilde{q}_{t_2}(\mathbf{u}_{t_1}) = \int p_{t_2}(\mathbf{u}_{t_1}|\mathbf{u}_{t_2})q_{t_2}(\mathbf{u}_{t_2})\mathrm{d}\mathbf{u}_{t_2}$

6: $\tilde{\mathbf{S}}_{t_1 t_2} = \mathbf{K}^{t_2}_{\mathbf{u}_{t_1}\mathbf{u}_{t_1}} - \mathbf{K}^{t_2}_{\mathbf{u}_{t_1}\mathbf{u}_{t_2}}(\mathbf{K}^{t_2}_{\mathbf{u}_{t_2}\mathbf{u}_{t_2}})^{-1}\mathbf{K}_{\mathbf{u}_{t_2}\mathbf{u}_{t_1}} + \mathbf{K}^{t_2}_{\mathbf{u}_{t_1}\mathbf{u}_{t_2}}\mathbf{K}^{t_2}_{\mathbf{u}_{t_2}\mathbf{u}_{t_2}}\mathbf{S}_{u_{t_2}}(\mathbf{K}^{t_2}_{\mathbf{u}_{t_2}\mathbf{u}_{t_2}})^{\intercal}\mathbf{K}^{t_2}_{\mathbf{u}_{t_2}\mathbf{u}_{t_1}}$
▷ Covariance of $\tilde{q}_{t_2}(\mathbf{u}_{t_1}) = \int p_{t_2}(\mathbf{u}_{t_1}|\mathbf{u}_{t_2})q_{t_2}(\mathbf{u}_{t_2})\mathrm{d}\mathbf{u}_{t_2}$

7: $\text{KL} \leftarrow \text{KL}(\mathcal{N}(\mathbf{m}_{\mathbf{u}_{t_2}}, \mathbf{S}_{\mathbf{u}_{t_2}})||\mathcal{N}(0, \mathbf{K}^{t_2}_{\mathbf{u}_{t_2}\mathbf{u}_{t_2}}))$

8:
$$\text{CorrectionTerm}(t_1, t_2) \leftarrow \text{KL}(\mathcal{N}(\tilde{\mathbf{m}}_{t_1 t_2}, \tilde{\mathbf{S}}_{t_1 t_2})||\mathcal{N}(0, \mathbf{K}^{t_1}_{\mathbf{u}_{t_1}\mathbf{u}_{t_1}}))$$
$$- \text{KL}(\mathcal{N}(\tilde{\mathbf{m}}_{t_1 t_2}, \tilde{\mathbf{S}}_{t_1 t_2})||\mathcal{N}(\mathbf{m}_{\mathbf{u}_{t_1}}, \mathbf{S}_{\mathbf{u}_{t_1}}))$$

9: **return** $\ell_{\text{varexp}} - \text{KL} + \text{CorrectionTerm}(t_1, t_2)$

# F  Additional results

## F.1  Full results including RMSE and ECE

We also include the full results of test NLPD and RMSE (ECE for experiments on COVID due to non-Gaussian likelihood) containing evaluation of Task $i$ after learning tasks $j = i, i + 1, \cdots, 10$ (5 for experiments on COVID) for all $i$.

We define the Root Mean Squared Error (RMSE) and Negative Log Predictive Density (NLPD) as the following:

$$\text{RMSE} = \sqrt{\frac{1}{N}\sum_{i=1}^{N}(y_i - \hat{y}_i)^2}, \quad \text{NLPD} = -\frac{1}{N}\sum_{i=1}^{N}\log \hat{p}(y_i \mid x_i). \tag{40}$$

$$\tag{41}$$

The Expected Calibration Error ([Guo et al., 2017]; ECE) is defined as

$$\text{ECE} = \frac{1}{K}\sum_{k=1}^{K}\left|\frac{1}{N}\sum_{i=1}^{N}\mathbf{1}\big(y_i \in \big[\hat{q}_{\frac{1-c_k}{2}}(x_i), \hat{q}_{\frac{1+c_k}{2}}(x_i)\big]\big) - c_k,\right| \tag{42}$$

where

$$K = 10, \quad c_k \in \{0.05,\ 0.15,\ \ldots,\ 0.95\}$$

$N$ is the number of test points $(x_i, y_i)$,

$\hat{q}_p(x_i) =$ the empirical $p$–quantile of the $S$ predictive samples $\{\hat{y}_i^{(s)}\}_{s=1}^{S}, \quad S = 100$

$$\mathbf{1}(\cdot) = \begin{cases} 1, & \text{if the argument is true,} \\ 0, & \text{otherwise.} \end{cases}$$

Figure 7 shows the RMSE results for time series regression experiments (results of NLPD are in Figure 2 in the main text). Figure 8 and 9 show NLPD and ECE results on COVID dataset with an additional sanity check baseline described in Appendix D.1. Figure 10 - 13 show full results of RMSE and NLPD for UCI datasets, and Figure 14 and 15 show full results of RMSE and NLPD for ERA5 dataset.

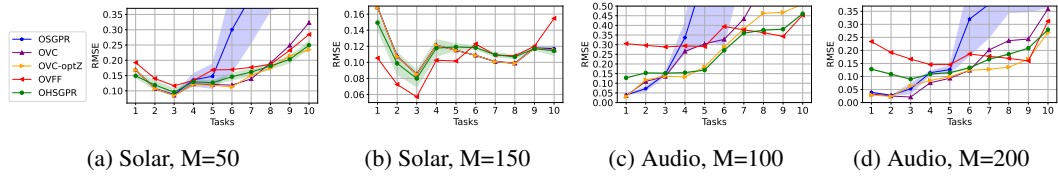

    (a) Solar, M=50          (b) Solar, M=150         (c) Audio, M=100       (d) Audio, M=200

Figure 7: Test set RMSE over the learned tasks vs. number of learned tasks for Solar Irradiance and Audio signal prediction dataset.

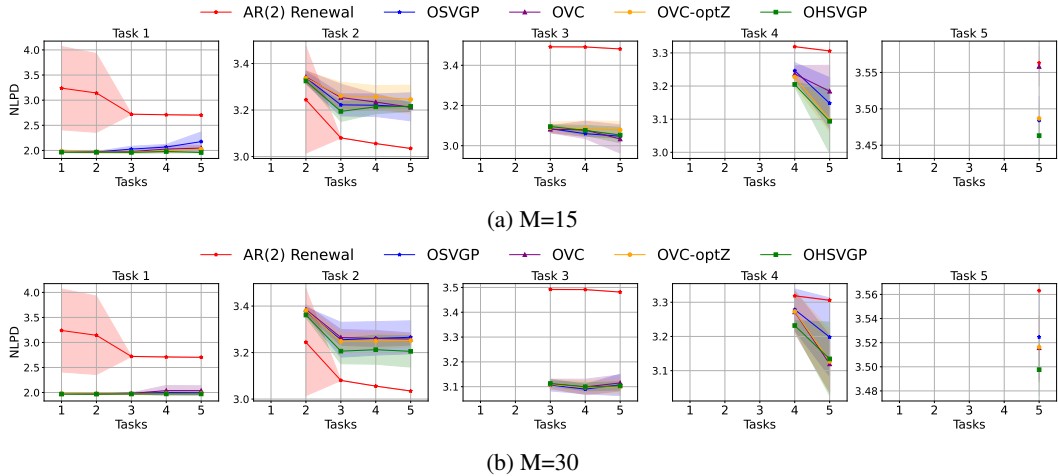

(a) M=15

(b) M=30

Figure 8: Test set NLPD per task after continually learning each task for all the 5 tasks on COVID dataset.

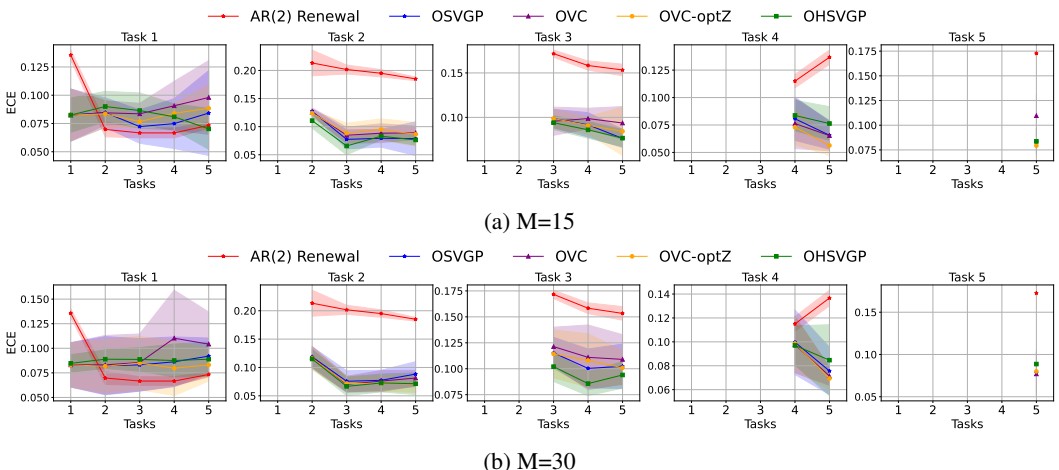

Figure 9: Test set ECE per task after continually learning each task for all the 5 tasks on COVID dataset.

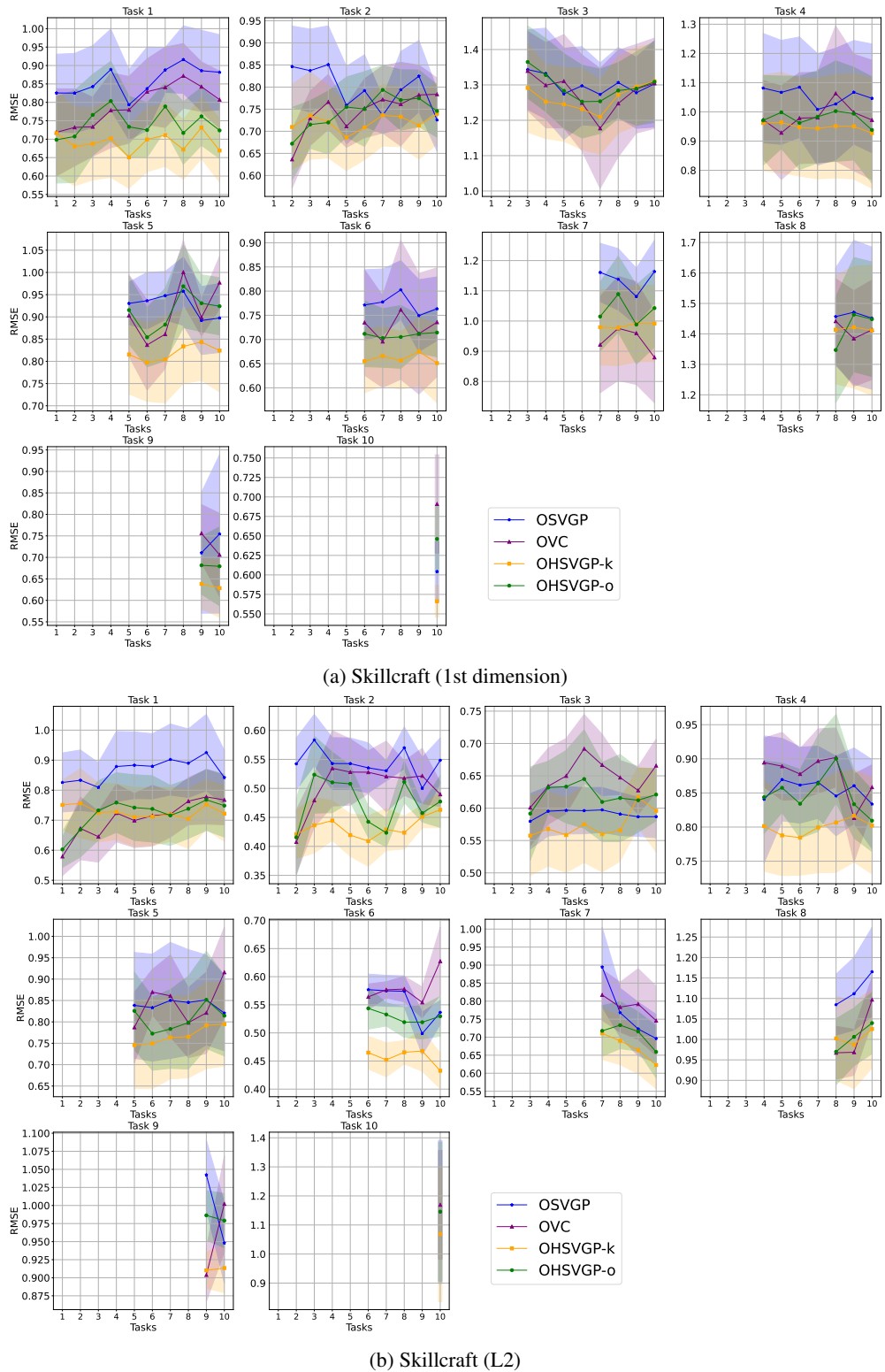

(a) Skillcraft (1st dimension)

(b) Skillcraft (L2)

Figure 10: Test set RMSE per task after continually learning each task for all the 10 tasks on UCI Skillcraft dataset.

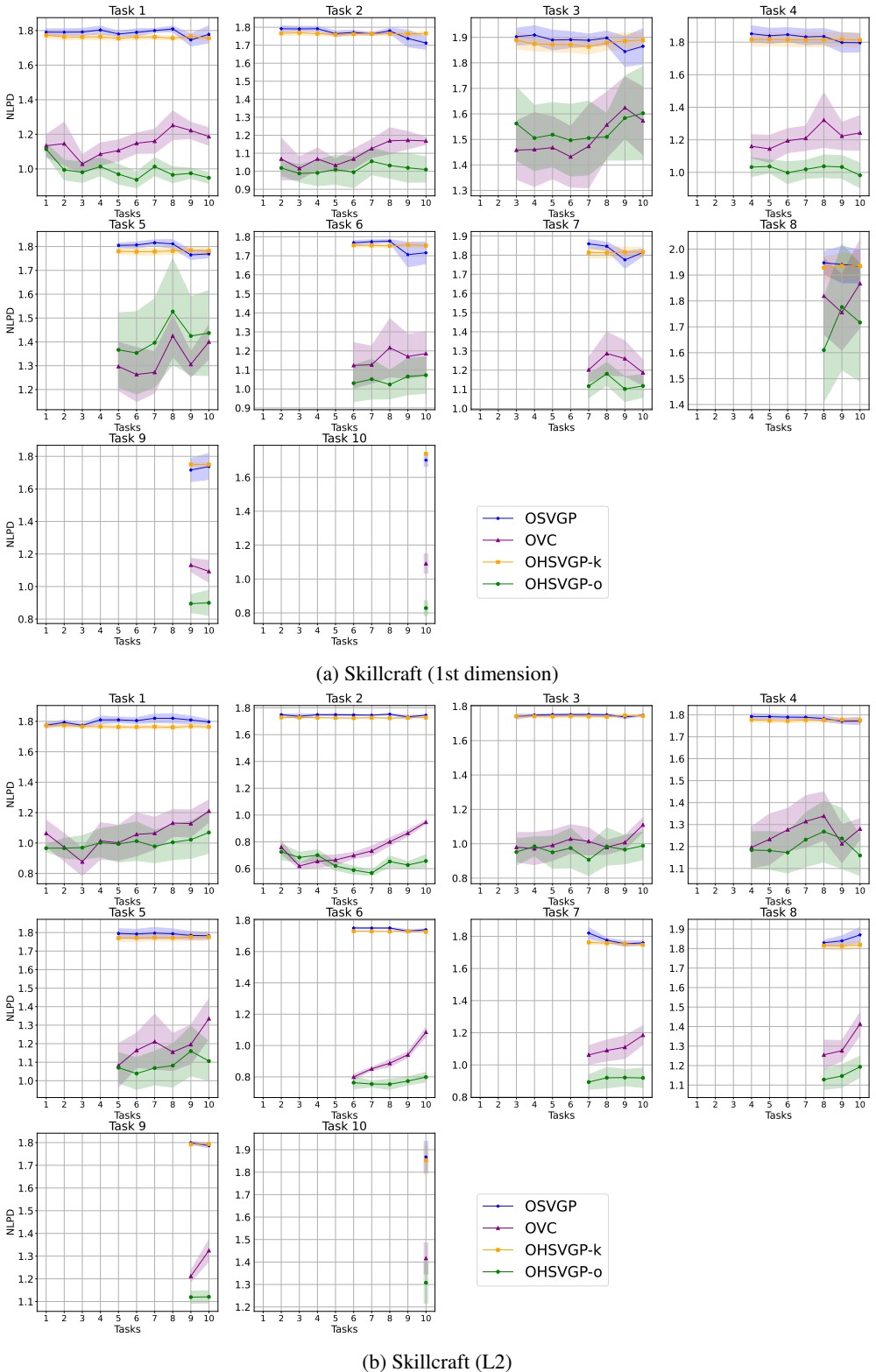

(a) Skillcraft (1st dimension)

(b) Skillcraft (L2)

Figure 11: Test set NLPD per task after continually learning each task for all the 10 tasks on UCI Skillcraft dataset.

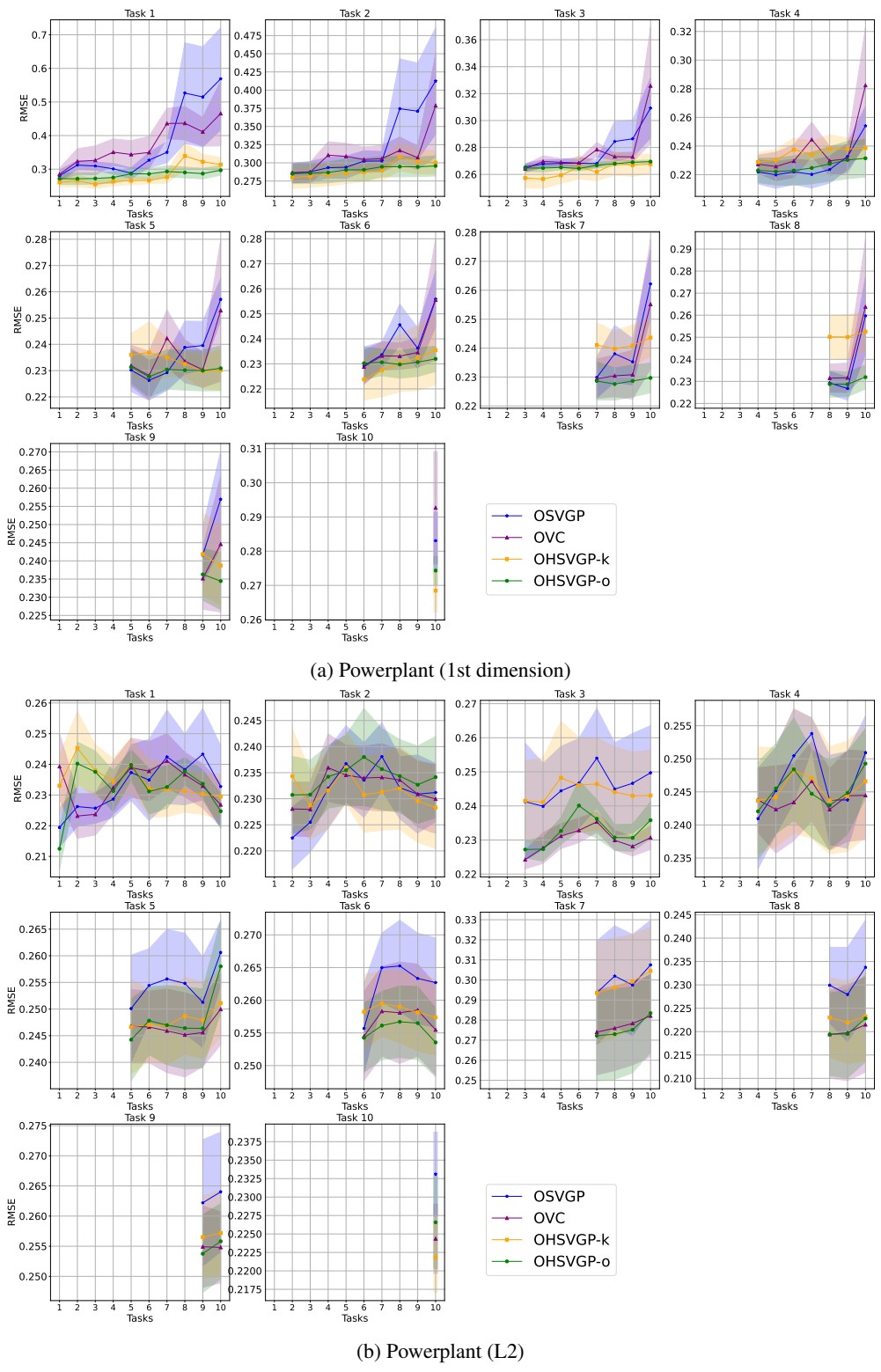

(a) Powerplant (1st dimension)

(b) Powerplant (L2)

Figure 12: Test set RMSE per task after continually learning each task for all the 10 tasks on UCI Powerplant dataset.

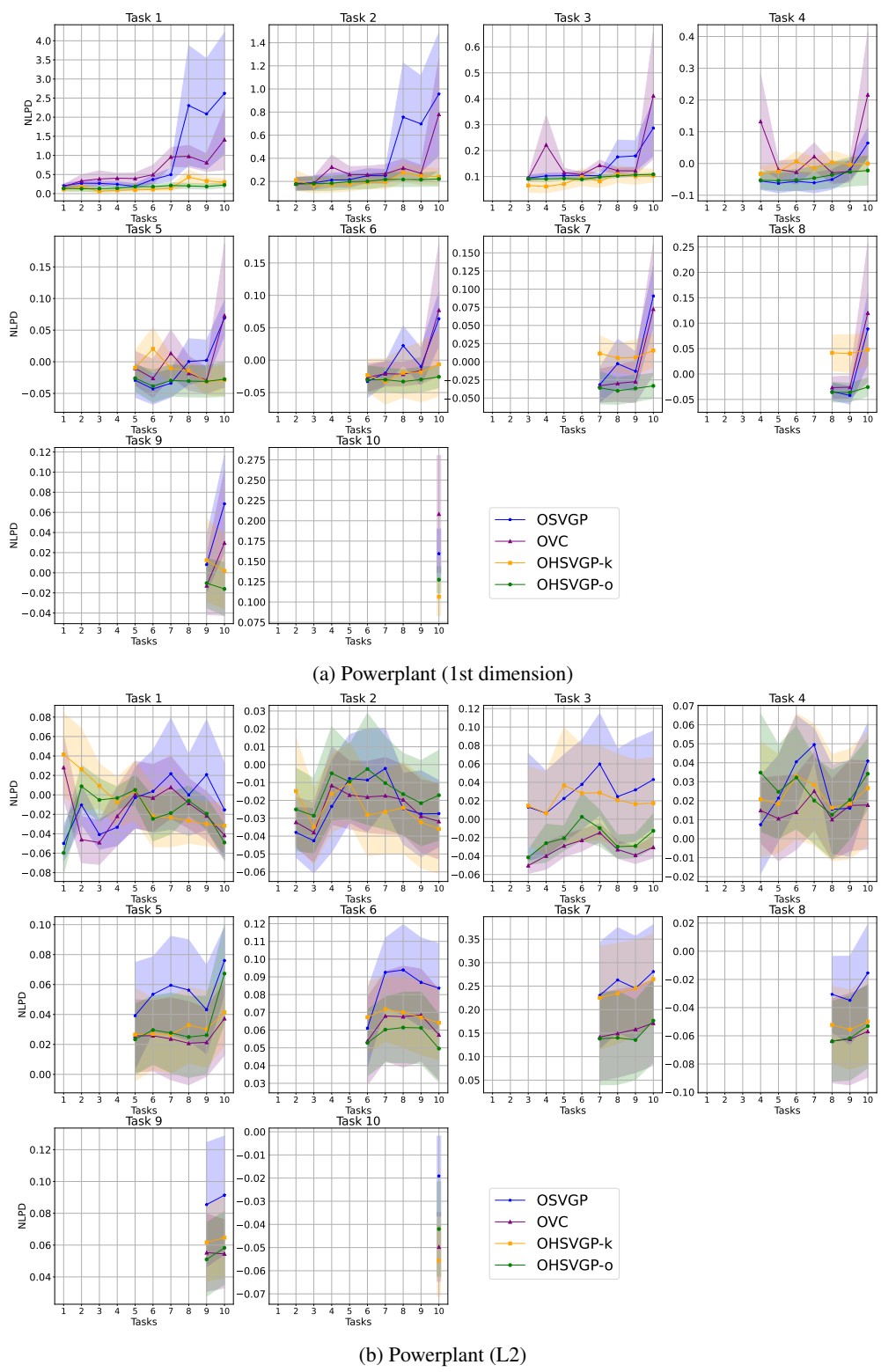

(a) Powerplant (1st dimension)

(b) Powerplant (L2)

Figure 13: Test set NLPD per task after continually learning each task for all the 10 tasks on UCI Powerplant dataset.

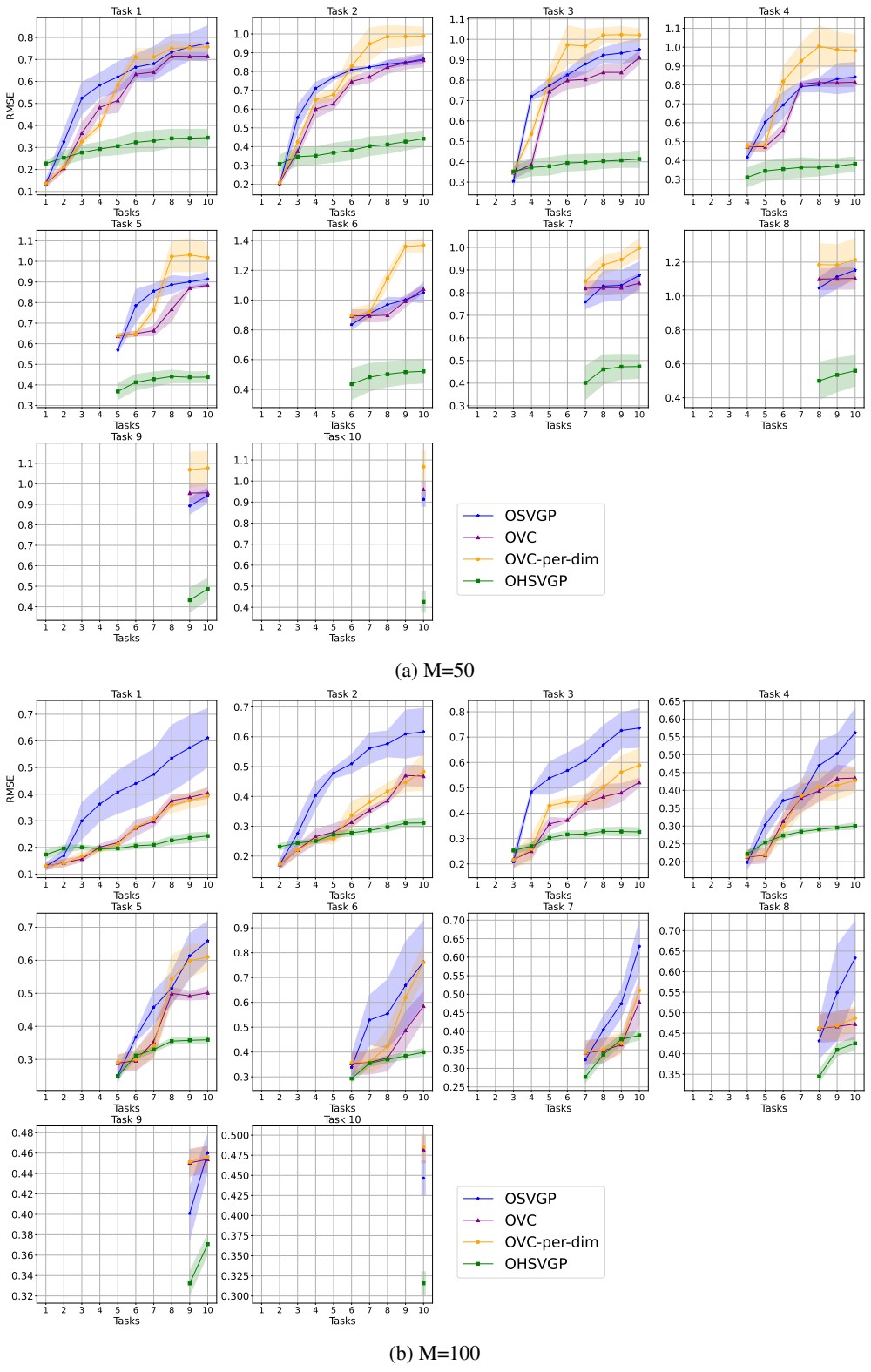

(a) M=50

(b) M=100

Figure 14: Test set RMSE per task after continually learning each task for all the 10 tasks on ERA5 dataset.

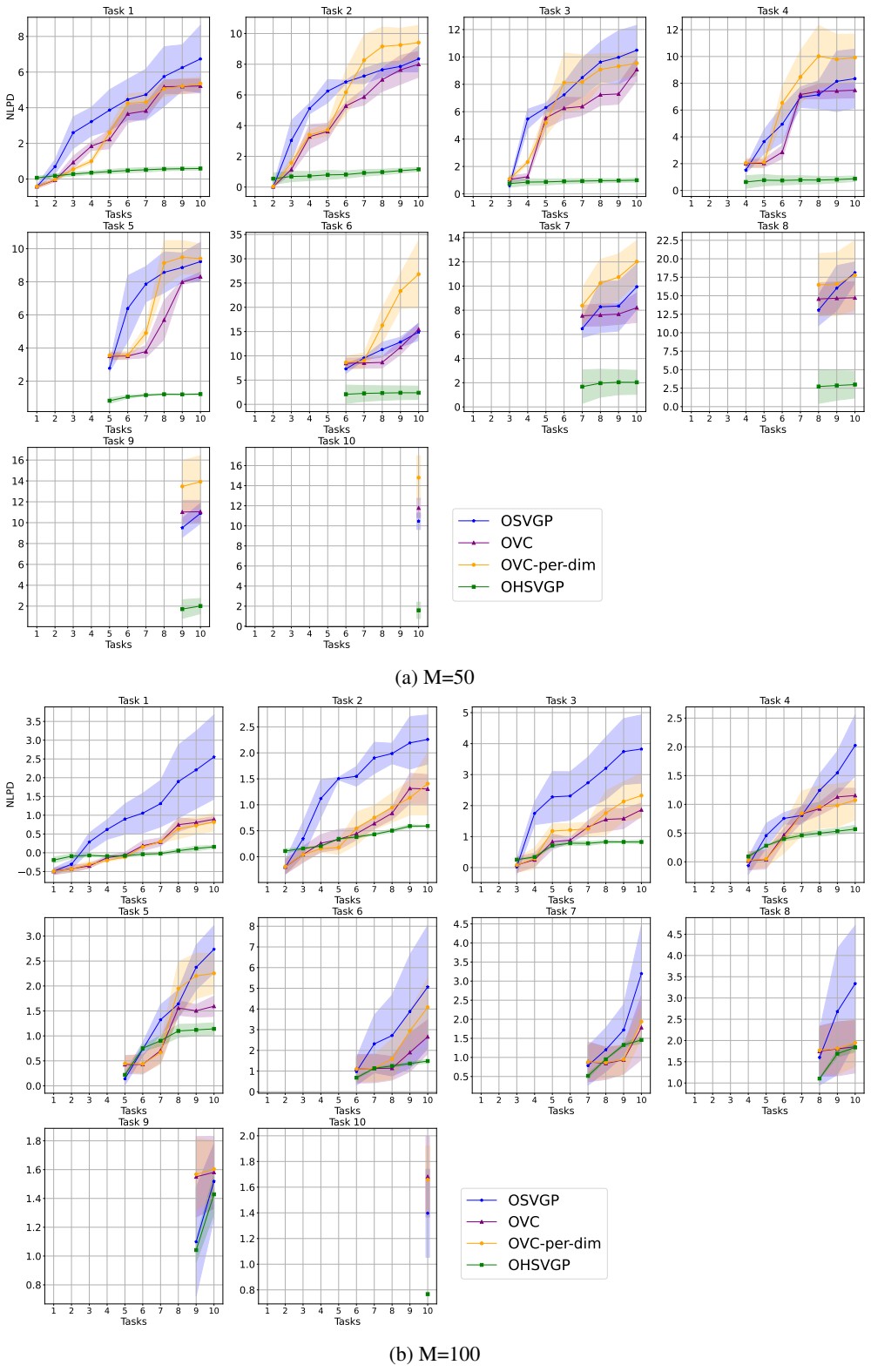

(a) M=50

(b) M=100

Figure 15: Test set NLPD per task after continually learning each task for all the 10 tasks on ERA5 dataset.

## F.2 Results for time series regression with trainable kernel hyperparameters

Figure 16 and 17 show RMSE and NLPD results for time series regression experiments based on trainable kernel hyperparameters (i.e., keep optimizing kernel hyperparameters online in all the tasks). Notice that OVC is only compatible with fixed kernel [Maddox et al., 2021], so we don't consider it here. Compared with the results based on fixed kernel in the main text, here all methods show less stable performance. Previous works either find a well-performed fixed kernel [Maddox et al., 2021] or scale the KL terms in the online ELBO objective with a positive factor requiring careful tuning to mitigate the unstable online optimization of kernel hyperparameters [Stanton et al., 2021, Kapoor et al., 2021].

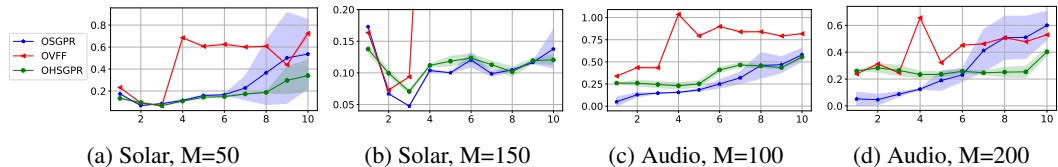

(a) Solar, M=50      (b) Solar, M=150      (c) Audio, M=100      (d) Audio, M=200

Figure 16: Test set RMSE over the learned tasks vs. number of learned tasks for Solar Irradiance and Audio signal prediction dataset (keep updating kernel hyperparameters).

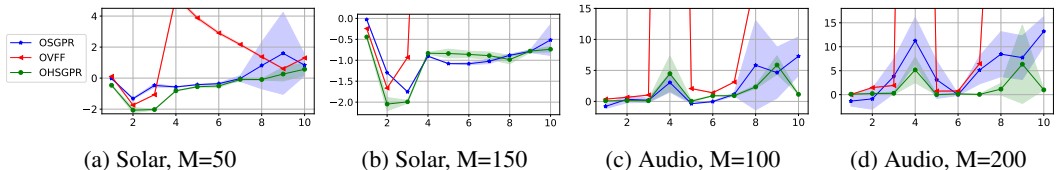

(a) Solar, M=50      (b) Solar, M=150      (c) Audio, M=100      (d) Audio, M=200

Figure 17: Test set NLPD over the learned tasks vs. number of learned tasks for Solar Irradiance and Audio signal prediction dataset (keep updating kernel hyperparameters).

## F.3 Visualization of impacts of sorting criterion for OHSVGP in continual learning

Here, we consider fitting OHSVGPs with 20 inducing variables for a continual binary classification problem on the Two-moon dataset [Ganin et al., 2016]. The data is splitted into three task and we use a Bernoulli likelihood to model binary labels. We consider three different sorting criteria:

- **Random,** denoted as OHSVGP-rand. The order of data points in each task is obtained via random permutation.

- **Kernel similarity maximization,** denoted as OHSVGP-k-max. We select the $i$-th point in task $j$ to be $\boldsymbol{x}_i^{(j)} = \arg\max_{\boldsymbol{x} \in \boldsymbol{X}^{(j)}} k(\boldsymbol{x}, \boldsymbol{x}_{i-1}^{(j)})$ for $i > 1$, and the first point in first task is set to be $\boldsymbol{x}_1^{(1)} = \arg\max_{\boldsymbol{x} \in \boldsymbol{X}^{(1)}} k(\boldsymbol{x}, \boldsymbol{0})$. The intuition is that the signals to memorize, when computing the prior covariance matrices, tend to be more smooth if the consecutive $\boldsymbol{x}$'s are close to each other.

- **Kernel similarity minimization,** denoted as OHSVGP-k-min. We select the $i$-th point in task $j$ to be $\boldsymbol{x}_i^{(j)} = \arg\min_{\boldsymbol{x} \in \boldsymbol{X}^{(j)}} k(\boldsymbol{x}, \boldsymbol{x}_{i-1}^{(j)})$ for $i > 1$, and the first point in first task is set to be $\boldsymbol{x}_1^{(1)} = \arg\min_{\boldsymbol{x} \in \boldsymbol{X}^{(1)}} k(\boldsymbol{x}, \boldsymbol{0})$. In this case, we deliberately make it difficult to memorize the signals in the recurrent computation for the prior covariance matrices.

Figure 18 show the decision boundaries after each task for different OHSVGPs based on different sorting criteria. We also include the decision boundaries of an OSVGP model for reference. Both OHSVGP-k-max and OHSVGP-rand return decision boundaries achieving 100% accuracy, while OHSVGP-k-min show catastrophic forgetting after Task 3, which suggests OHSVGP requires a sensible sorting criterion to perform well in continual learning tasks.

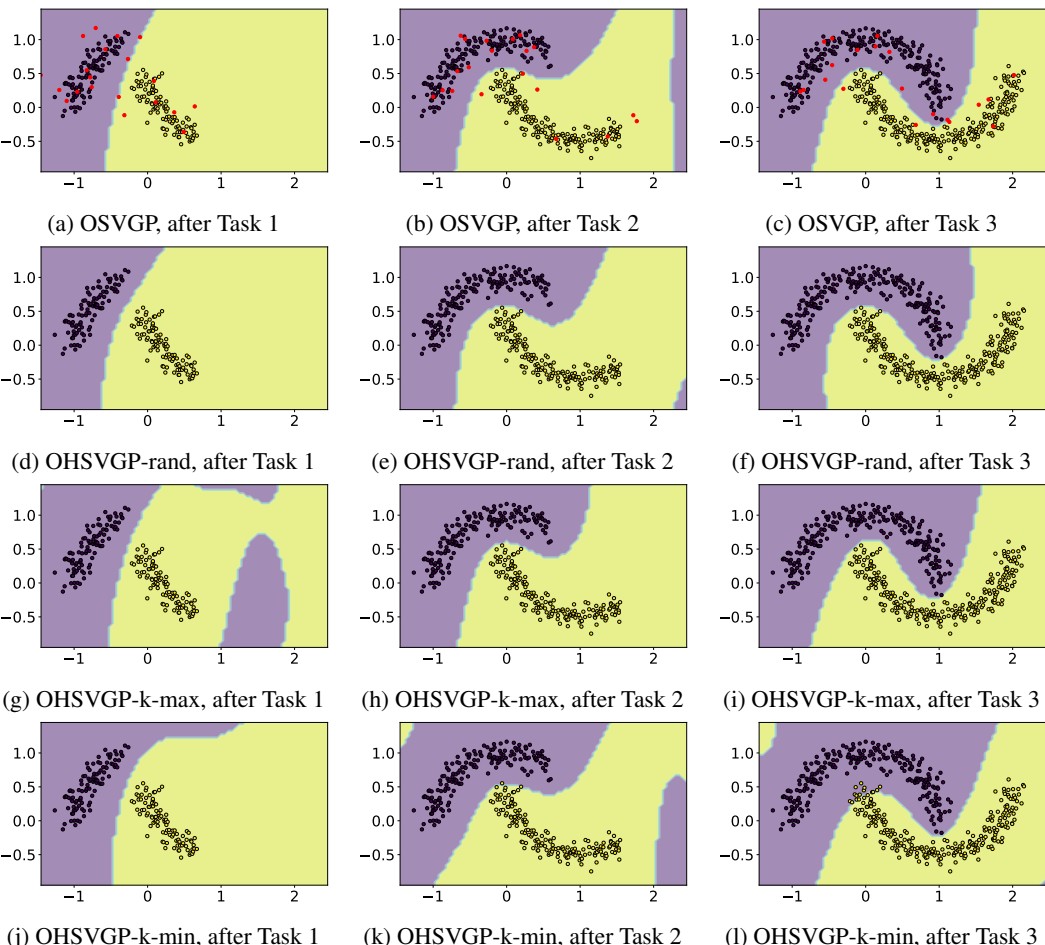

Figure 18: Decision boundaries of OSVGP, and OHSVGP models with different sorting criteria after each task (3 in total) on the Two-moon dataset. For OSVGP, we visualize the inducing points with red color.

### F.4 Comparison of basis–measure variants

Figure 19 shows the results of OHSGPR applied to a toy time-series regression dataset, where the data is split chronologically into three equal segments, used as Tasks 1–3 in an online learning setup. The figure compares the effect of several variants of the HiPPO operators [Gu et al., 2020, 2023] when used for OHSGPR. Subfigures (a–c) correspond to HiPPO-LegS as used in all of our main experiments. Subfigures (d–f) apply HiPPO-LegT, (g–i) apply HiPPO-LagT, based on the Laguerre polynomial basis, and (j–l) apply HiPPO-FouT, based on Fourier basis functions. While OHSGPR-LegS successfully memorizes all the past tasks, OHSGPR-LegT, OHSGPR-LagT and OHSGPR-FouT all demonstrate catastrophic forgetting to certain degree since instead of the uniform measure over the past (as is used in HiPPO-LegS), they are based on measures which place more mass over the recent history. LegT and FouT use a fixed-length sliding window measure, while LagT uses exponentially decaying measure, which assigns more importance to recent history.

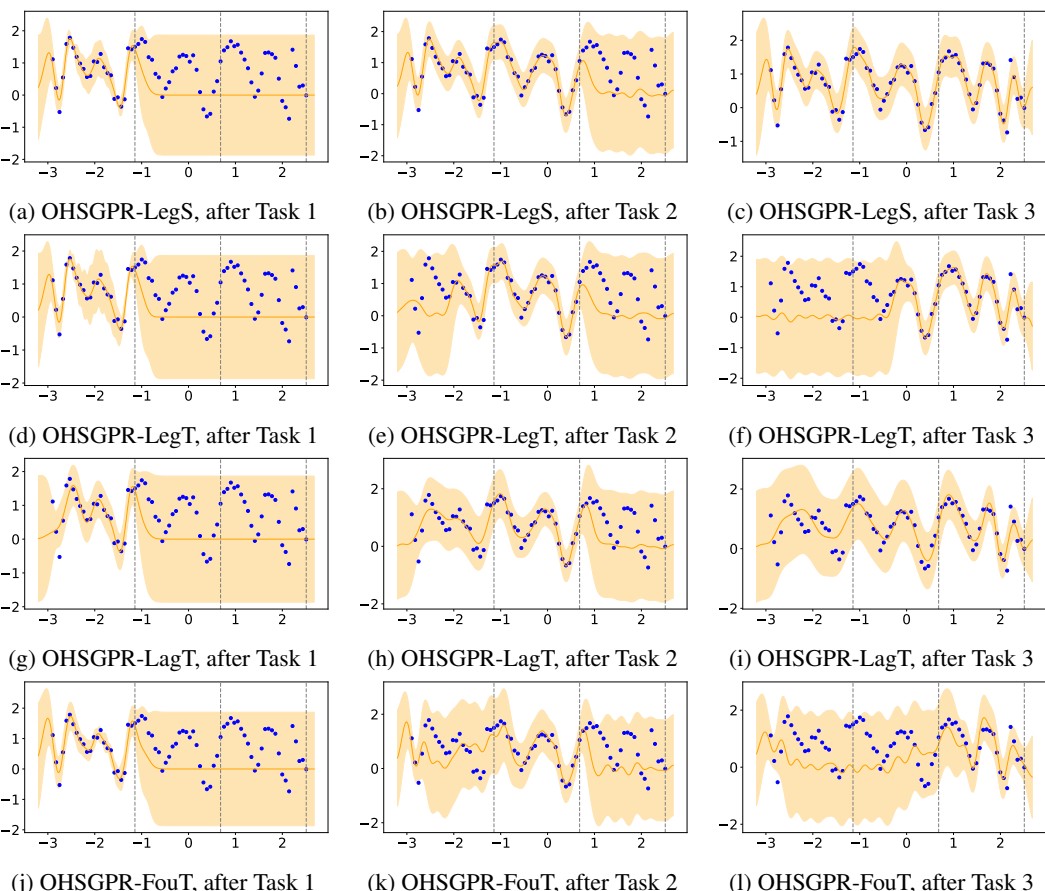

Figure 19: Comparison of OHSGPR based on different HiPPO variants on a toy online regression dataset.

