# OpenReview forum: "Recurrent Memory for Online Interdomain Gaussian Processes"
_NeurIPS.cc/2025/Conference — NeurIPS 2025 poster_

### Official Review · Reviewer_Aht3 · 2025-06-19

**Clarity:** 3
**Significance:** 4
**Originality:** 4
**Rating:** 5
**Confidence:** 3

**Summary:**

The authors propose the OHSVGP model, which marries the HiPPO and the interdomain GP frameworks to allow online learning with long-range memory support. The paper also indicate how the kernel matrices can be updated recurrently by handling the ODE involved in the learning procedure. The performed experiments, in different scenarios, show promising results, compared to other methods from the literature.

**Questions:**

- Section 5.2 - Line 319: The heuristic sorting method for OHSVGP-k considers the minimization of k(x, x_{i-1}^{(j)}). Wouldn't it result in the "farthest" point, since the kernel is inversely proportional to the distance between the inputs? It seems that if the intent is to find the closest point, one should maximize the kernel value between the points. Appendix E.3 also covers this point.

- How exactly the "High dimensional time series prediction" task was conducted with GPVAEs in the experiments? The description in Section 5.3 focus on the hyperparameters of the model components, but does not detail the overall setup.

- Line 99: Wrong reference for the original VAE paper.

**Ethical Concerns:**

["NO or VERY MINOR ethics concerns only"]

**Final Justification:**

The authors addressed my points and I maintain my favorable score.

**Limitations:**

The authors state the limitations of their work.

**Paper Formatting Concerns:**

None.

**Quality:**

4

**Strengths And Weaknesses:**

The paper is well organized and written, with all the necessary background properly summarized for the reader. In that regard, both the related work and the baseline comparisons are also well covered. The experimental section is detailed and transparent in terms of settings and hyperparameters choices. I also praise the authors for the provided comprehensive supplementary material, which contains several interesting additional descriptions and experiments.

The main weak point might be the quantity of steps and decisions involved in the training of the proposed approach, with some particularities for each experimental scenario. A complete algorithm, summarizing all the calculations and methodological decisions in each case, would be great for better reader understanding.

---

> ### Author Rebuttal · Authors · 2025-07-30
>
> ### Global Response:
>
> We sincerely thank all reviewers for the constructive feedbacks and the overall positive reviews. We are encouraged that reviewers appreciated our **novel (CC7b, GFdF, Aht3)** and **technically sound (D83n, GFdF)** method that addresses the **critical (D83n)** catastrophic forgetting problem in online GPs. In general, the reviewers are happy with the **clarity (D83n, Aht3)** of the manuscript and our **comprehensive (CC7b, HjVk) experimental evaluation** of OHSVGP, which shows **strong performance (D83n, HjVk, GFdF, Aht3)** compared with baselines.
>
> ---
> ### Individual Response to Reviewer Aht3:
>
> Below we address the questions from Reviewer Aht3.
>
> >Q1. A complete algorithm, summarizing all the calculations and methodological decisions in each case.
>
> We agree that a complete algorithm will improve the manuscript and we will include it in future revision of the paper. For now, we give a brief sketch of the algorithm in the cases with conjugate Gaussian likelihood (time series regression) and with non-conjugate likelihoods (all the other experiments):
>
> * Case 1 (Gaussian likelihood). Assume we have the old kernel matrices ($K_{fu}^{old}$ and $K_{uu}^{old}$) and optimal variational parameters ($m_{old}$ and $S_{old}$) computed from the last task.
>     * Update $K_{fu}^{old}$ and $K_{uu}^{old}$ to $K_{fu}^{new}$ and $K_{uu}^{new}$, respectively, based on HiPPO recurrences to cover the time region of the current task (Eq. 4 and Appendix B).
>     * Since Gaussian likelihood is conjugate to GP prior, the optimal $m_{new}$ and $S_{new}$ are analytically tractable and can directly be computed based on $m_{old}$, $S_{old}$, and these kernel matrices without training (detailed formulas can be found in Appendix B in [1]).
>     * Construct the new posterior predictive based on $m_{new}$, $S_{new}$, and kernel matrices.
>
> * Case 2 (non-conjugate likelihood). Assume we have the old kernel matrices ($K_{fu}^{old}$ and $K_{uu}^{old}$) and variational parameters ($m_{old}$ and $S_{old}$) obtained from the last task.
>     * Update $K_{fu}^{old}$ and $K_{uu}^{old}$ to $K_{fu}^{new}$ and $K_{uu}^{new}$, respectively, based on HiPPO recurrences to cover the time region of the current task (Eq. 4 and Appendix B).
>     * Construct the online ELBO (Eq. 3), based on $m_{old}$, $S_{old}$, and kernel matrices, as training objective for $m_{new}$ and $S_{new}$, and train $m_{new}$ and $S_{new}$ by gradient-based optimization of the online ELBO.
>     * Construct the new posterior predictive based on $m_{new}$, $S_{new}$, and kernel matrices.
>
> [1] "Streaming Sparse Gaussian Process Approximations", NeurIPS 2017
>
> > Q2. Clarification of OHSVGP-k.
>
> You are right. **This is a typo - OHSVGP-k actually considers $ \argmax_x k(x, x_{i-1})$ (the results of OHSVGP-k-min and OHSVGP-k-max in Appendix E.3 should also be swapped).** We double-checked that our implementation of OHSVGP-k in facts considers $\argmin_x \sum_{d=1}^D \frac{(x^{(d)} - x_{i-1}^{(d)})^2}{l_d^2}$ (where $D$ is the dimension of $x$ and $l_d$'s are per-diemension length-scales), which is equivalent to the maximization of the kernel values. Thank you for pointing it out and we will fix this typo in our next revision of the manuscript.
>
> > Q3. Background and experimental Setups of GPVAEs.
>
> In GPVAE experiment, each data point is a multi-output/high-dimensional time series and GPVAE uses amortized inference based on an encoder to output an approximate GP posterior for each time series. We include the model specification of GPVAE used in this work in Appendix D.2 and we will include a pointer to it in the main text in our next revision of the manuscript. The dataset of time series is splitted into multiple tasks and the continual learning is achieved by (i) imposing an additional EWC loss to the encoder and decoder networks and (ii) updating the kernel matrices via HiPPO recurrence to keep adapting to the new time region associated with the time series in the new task.
>
> > Q4. Wrong reference of VAE paper.
>
> Thank you for pointing out the typo. We will fix it in our next revision of the manuscript.

---

> > ### Comment · Reviewer_Aht3 · 2025-08-02
> > **Thanks for the comments**
> >
> > The authors addressed my points and I maintain my favorable score.

---

### Official Review · Reviewer_GFdF · 2025-06-29

**Clarity:** 4
**Significance:** 3
**Originality:** 3
**Rating:** 4
**Confidence:** 4

**Summary:**

This paper proposes OHSVGP, an online Gaussian Process model that integrates the HiPPO framework into sparse variational GPs (SVGP) to address forgetting in continual learning. This allows recurrent online updates of kernel matrices via HiPPO's ODE structure, enabling long-term memory compression. Results show good predictive performance, memory retention, and computational efficiency compared to OSVGP, OVC, and OVFF baselines.

**Questions:**

What is the convergence behavior and computational cost of using RFF to approximate Kuu w.r.t. the number of RFF samples?
Furthermore, have you conducted a time complexity analysis to facilitate a general comparison with other baseline models?
Since your work incorporates both RFF-based and inducing point-based approximations, have you considered comparing it with RFF-based online Gaussian Processes (SSM)?

**Ethical Concerns:**

["NO or VERY MINOR ethics concerns only"]

**Limitations:**

Yes, and see the weakness.

**Quality:**

4

**Strengths And Weaknesses:**

Strengths: Good work to bridge HiPPO (SSM-based memory) with interdomain GPs for online learning. The reformulation of HiPPO projections as time-dependent inducing variables is novel and theoretically grounded.

Weaknesses: Performance drops significantly with heuristic sorting (OHSVGP-k vs. OHSVGP-o). This reliance on "oracle" task ordering limits applicability to real-world continual learning.

---

> ### Author Rebuttal · Authors · 2025-07-30
>
> ### Global Response:
>
> We sincerely thank all reviewers for the constructive feedbacks and the overall positive reviews. We are encouraged that reviewers appreciated our **novel (CC7b, GFdF, Aht3)** and **technically sound (D83n, GFdF)** method that addresses the **critical (D83n)** catastrophic forgetting problem in online GPs. In general, the reviewers are happy with the **clarity (D83n, Aht3)** of the manuscript and our **comprehensive (CC7b, HjVk) experimental evaluation** of OHSVGP, which shows **strong performance (D83n, HjVk, GFdF, Aht3)** compared with baselines.
>
> ---
>
> ### Individual Response to Reviewer GFdF:
>
> Below we address the questions from Reviewer GFdF.
>
> > Q1. Performance drops with OHSVGP-k (heuristic sorting) in UCI experiments.
>
> For continual learning problems where the order of data points within each task is unknown, the sorting method will impact the performance of OHSVGP and what sorting method may be preferable remains an open question. OHSVGP-k, which sorts the data points based on the kernel distances among them, indeed achieves worse performance, than OHSVGP-o, based on oracle ordering. However, OHSVGP-k is still decently robust in the sense that it achieves competitive or better results than the OSVGP baseline as shown in Figure 5.
>
> This sorting step also can not be avoided in deep SSMs, such as Mamba (advanced extension of HiPPO), when applied to data modalities without pre-defined ordering (e.g., Vision [1], where Mamba approach requires the user to specify an ordering of the patches split from an image). Empirically, some heuristic sorting strategies can still enable good performance of deep SSMs on these unordered data.
>
> [1] "Vision Mamba: Efficient visual representation learning with bidirectional state space model", ICML 2024
>
> > Q2. RFF approximation quality and computational cost.
>
> We expect there is a tradeoff between computational cost due to RFF sample size and the performance. However, with the current number of RFF samples considered in our experiments, we are already able to both obtain better results compared with other baselines and also reduce the training time.
>
> OHSVGP and OVC are almost equally fast and both of them are significantly more efficient than OSVGP. We report the time metric for time series regression experiments in Table 1. For continual learning on UCI datasets, the training time of OHSVGP and OVC are about half compared with OSVGP. The time complexity of OHSVGP and OVC are the same after obtaining the kernel matrices: $O(M^3 + N^2M)$, where M is the number of inducing points and N is the number of data points in the current task. Both OVC and OHSVGP update kernel matrices just once before entering the training loops every time when a new task arrives. OHSVGP further requires an additional step to sort the data points in the task for continual learning. However, all these one-time costs are negligible compared with the training loops for the variational parameters, and inducing locations (for OSVGP only).
>
> All the baselines considered in the experiments are not based on RFF but based on exact kernel matrices, so even RFF may introduce some approximation bias, the benefit of OHSVGP in terms of long-term memory completely outweights it. We can turn the baselines into RFF-based online Gaussian processes by also approximating kernel evaluations with RFF, but we believe it is not likely to improve their performances. We are not aware of online GP methods tailored to RFF approximations and we would be interested if you can provide some references.

---

> > ### Comment · Reviewer_GFdF · 2025-08-07
> >
> > Thanks for the response. I'll keep my score but would like to see if you could consider the online RFF as baseline in future.
> >
> > [1] "Sequential estimation of Gaussian process-based deep state-space models." IEEE Transactions on Signal Processing

---

### Official Review · Reviewer_HjVk · 2025-06-30

**Clarity:** 4
**Significance:** 3
**Originality:** 3
**Rating:** 5
**Confidence:** 4

**Summary:**

This paper incorporates the Hippo framework within the interdomain GP for developing an online novel GP method that can memory the history information.

**Questions:**

Some minor comments:

In the background section, the introduction of GPVAE appears somewhat unnecessary, as it is not essential to the main narrative. Moreover, the summary at the beginning of the section does not mention GPVAE, which makes its inclusion feel somewhat disconnected.

**Ethical Concerns:**

["NO or VERY MINOR ethics concerns only"]

**Final Justification:**

The author addressed my concern and I will keep my positive score.

**Limitations:**

See the weaknesses and questions part.

**Quality:**

3

**Strengths And Weaknesses:**

Strengths:

1. Establishes a connection between the Hippo framework and inducing points from the perspective of interdomain GPs.

2. Leverages RFF to decouple the two inputs, enabling the ODE-based evolution of Kuu.

3. Provides comprehensive simulations that demonstrate not only the model’s superior performance on various tasks but also its computational efficiency.

4. The paper is clearly written and easy to follow.

Weaknesses:
Overall, the paper is well-executed. While the connection between the Hippo framework and interdomain GP is clearly presented, it may be seen as a natural extension rather than a fundamentally novel contribution, as it is quite straightforward.

---

> ### Author Rebuttal · Authors · 2025-07-30
>
> ### Global Response:
>
> We sincerely thank all reviewers for the constructive feedbacks and the overall positive reviews. We are encouraged that reviewers appreciated our **novel (CC7b, GFdF, Aht3)** and **technically sound (D83n, GFdF)** method that addresses the **critical (D83n)** catastrophic forgetting problem in online GPs. In general, the reviewers are happy with the **clarity (D83n, Aht3)** of the manuscript and our **comprehensive (CC7b, HjVk) experimental evaluation** of OHSVGP, which shows **strong performance (D83n, HjVk, GFdF, Aht3)** compared with baselines.
>
> ---
>
> ### Individual Response to Reviewer HjVk:
>
> Below we address the questions from Reviewer HjVk.
>
> > Q1. Clarification of the novelty and contribution.
>
> OHSVGP is indeed built on two well-established frameworks, HiPPO and online SVGP. However, the identification of the connection between the two frameworks from two different domains (deep sequence modeling and probabilistic machine learning) is non-trivial and requires in-depth understanding of both areas. The novelties and contributions of many interdomain GP methods are about finding good basis functions that enable desired properties in the approximate GP (e.g., reduced computational complexity [1,2]). In our case, we leverage the long-term memory preservation property of HiPPO framework to build an online GP free of severe catastrophic forgetting. To our knowledge, this is the first work that considers interdomain GP in online setup and OHSVGP is the first interdomain GP method that considers time-varing basis function which keeps memorizing the whole history and also adapting to the new task. As a result, it is naturally suitble for online learning.
>
> [1] "Variational Fourier Features for Gaussian Processe", JMLR 2018
>
> [2] "Sparse Gaussian Processes with Spherical Harmonic Features", ICML 2020
>
> > Q2. The introduction of GPVAE in the Background appears somewhat unnecessary.
>
> We consider continual learning in GPVAE to show that OHSVGP can be scaled up to higher dimensional problems, which will be of interest to the wider machine learning community. Hence, we also give a short description of GPVAE in the Background section and include its detailed model specification in Appendix D.2 for readers that are not familiar with it. It is indeed not a core part of our algorithm, but rather an add-on. We will mention GPVAE in the summary at the beginning of the Background section and add a pointer to Appendix D.2 in future revision of the manuscript.

---

> > ### Comment · Reviewer_HjVk · 2025-08-05
> >
> > The authors addressed my concern and I will maintain my score.

---

### Official Review · Reviewer_D83n · 2025-07-01

**Clarity:** 3
**Significance:** 3
**Originality:** 3
**Rating:** 4
**Confidence:** 3

**Summary:**

The paper proposes Online HiPPO Sparse Variational Gaussian Process (OHSVGP), a novel method for online Gaussian process that preserves long-term memory through recurrent updates. The key idea is to interpret the HiPPO framework—originally developed for state-space models—as defining interdomain inducing variables via time-varying orthogonal projections. This construction allows the model to capture history compactly and update kernel matrices via ODEs, avoiding expensive optimization over inducing points. The method is applied to time series prediction, continual learning, showing better accuracy and computational efficiency compared to existing approaches.

**Questions:**

* Do the authors expect this approach to work for non-stationary kernels beyond the RFF approximation used here? Would something like non-stationary Fourier features (e.g., [43]) be feasible?
* Kernel hyperparameters are fixed after the first task in most experiments. While this is understandable for stability, it limits the model’s flexibility in adapting to data shifts. Is there a principled way to integrate kernel hyperparameter updates into the recurrence framework?
* Can you comment on the sensitivity of the method to the number of inducing variables $M$?

**Ethical Concerns:**

["NO or VERY MINOR ethics concerns only"]

**Final Justification:**

I maintain a positive overall opinion. The rebuttal addressed the main concerns I raised. The main unresolved aspect is the lack of formal stability/error analysis for the HiPPO recurrence discretization, but given the strong empirical results and sound methodological design I confirm the borderline-accept score.

**Limitations:**

Nothing major to flag

**Quality:**

3

**Strengths And Weaknesses:**

This is a well-written and technically sound paper addressing a critical issue in online GP learning. The integration of HiPPO into interdomain SVGPs is both elegant and well-motivated, and the empirical results are strong. The ODE-based kernel updates combined with RFF approximations are especially good from a scalability standpoint.

Overall, the method is original, effective, and cleanly presented.

Some comments:

* The idea of viewing the HiPPO projection coefficients as time-evolving inducing variables is elegant. However, it assumes a time-ordered input sequence. In Section 3.3 (and 5.2), the authors mention a pseudo-time ordering heuristic for unordered data. It would be good to clarify how robust the method is to different sorting heuristics, including the overhead for sorting.
* The HiPPO basis function evolution introduces a dependency on accurate numerical integration over time. The recurrence is discretized (Euler or bilinear), but no error analysis or stability bounds are provided. This is important since the quality of the memory encoding depends on these dynamics.
* In Figure 6, increasing the number of inducing variables from $M = 50$ to $M = 100$ yields no clear performance improvement and occasionally degrades performance. This contradicts the common intuition that more inducing variables should improve approximation quality. An ablation or sensitivity analysis would help clarify the cause.

---

> ### Author Rebuttal · Authors · 2025-07-30
>
> ### Global Response:
>
> We sincerely thank all reviewers for the constructive feedbacks and the overall positive reviews. We are encouraged that reviewers appreciated our **novel (CC7b, GFdF, Aht3)** and **technically sound (D83n, GFdF)** method that addresses the **critical (D83n)** catastrophic forgetting problem in online GPs. In general, the reviewers are happy with the **clarity (D83n, Aht3)** of the manuscript and our **comprehensive (CC7b, HjVk) experimental evaluation** of OHSVGP, which shows **strong performance (D83n, HjVk, GFdF, Aht3)** compared with baselines.
>
> ---
>
> ### Individual Response to Reviewer D83n:
>
> Below we address the questions from Reviewer D83n.
>
> > Q1. How robust OHSVGP is to sorting methods and its computational overhead.
>
> The sorting is conducted only once every time when a new task arrives and its cost is negligible compared with the training loops. The performance of OHSVGP does depend on the sorting method and we include a 2D-input binary continual classification example in Appendix E.3 for visualization of the impacts of different sorting methods (there is a typo in the manuscript - the results of OHSVGP-k-max and OHSVGP-k-min there should be swapped. See our response to Q2 from Reviewer Aht3 for details). In that example, we found that OHSVGP is robust as long as we do not use an advesarially constructed sorting strategy. In our experiments, we found OHSVGP-k, which heuristically sorts the data points based on the kernel distances among them, indeed achieves worse performance, than OHSVGP-o, based on oracle ordering. However, it is still decently robust in the sense that it achieves competitive or better performance than the OSVGP baseline as shown in Figure 5.
>
> > Q2. Discretization error of HiPPO recurrence.
>
> While we do not have a formal stability bound yet, empirically we observe that the specification of the discretization step size depends on how smooth the time series is (e.g., for smooth time series (implying larger length-scale), the step size can be relataively large). In our experiments, we found that setting the step size to be within $2 \times \text{length-scale}$ is able to gaurantee reliable performance.
>
> > Q3. In Figure 6, increasing $M$ does not improve performance.
>
> Increasing $M$ from 50 to 100 does improve the performance in Figure 6. **The scales of the y-axis in Figure 6(a) and Figure 6(b) are different.** We choose to maintain different scales in y-axis because otherwise the results in Figure 6(b) will be too squeezy.
>
> > Q4. Do the authors expect this approach to work for non-stationary kernels.
>
> Yes, as long as there is an available feature approximation for a chosen kernel (e.g., [1] provides an example of feature approximation for non-stationary kernel).
>
> [1] "Spatial Mapping with Gaussian Processes and Nonstationary Fourier Features", Journal of Spatial Statistics 2018.
>
> > Q5. Fixed kernel hyperparamters may limit adaptation.
>
> The instability of online kernel hyperparameters update is a common problem in online SVGPs. Previous works either fix the kernel hyperparameters to ensure stable performance [2] or rely on additional tricks, such as generalized variational inference [3,4] or replay buffer that keeps a subset of past data for retraining to make the online variational inference closer to batch variational inference [5]. Here, we adopt the simpler strategy of fixing kernel hyperparameters since it already gives us good results for the benchmarks considered in this work and the long-term memory capability of OHSVGP, which is our main contribution, has been sufficiently demonstrated. Moreover, the aforementioned tricks of mitigating the unstable online kernel hyperparameters updates can all be easily integrated into OHSVGP since they are completely orthogonal and compatible to our key contribution.
>
> [2] "Conditioning Sparse Variational Gaussian Processes for Online Decision-making", NeurIPS 2021
>
> [3] "Kernel Interpolation for Scalable Online Gaussian Processes", AISTATS 2021
>
> [4] "Variational Auto-regressive Gaussian Processes for Continual Learning", ICML 2021
>
> [5] "Memory-based Dual Gaussian Processes for Sequential Learning", ICML 2023
>
> > Q6. Sensitivity of the method to the number of inducing variables $M$.
>
> Similar to standard SVGP, typically performance improves when $M$ increases (possibly with diminishing return though). In terms of memory preservation, we found that catastrophic forgetting in OSVGP tends to be more severe when $M$ is small (e.g., Figure 2(a) vs Figure 2(b)). This is possibly because when $M$ is small, the location of each inducing point needs to be representative, while when $M$ is relatively large, the inducing points are more likely to sufficiently cover the historical regions even though some of them are placed at suboptimal locations.

---

> > ### Comment · Reviewer_D83n · 2025-08-05
> >
> > Thanks for the response. I keep a positive opinion on the paper.

---

### Official Review · Reviewer_CC7b · 2025-07-01

**Clarity:** 2
**Significance:** 2
**Originality:** 2
**Rating:** 4
**Confidence:** 3

**Summary:**

The paper proposes a new GP model based on interdomain GP and HiPPO. Inspired by the usage of HiPPO in approximating time series in RNN, the authors adopt the bases in HiPPO for online GP. They show how the corresponding kernel matrices can be updated with differential equations (which is later approximated by difference equations). In the experiments, the proposed algorithm is evaluated in several problems (including online prediction, continual learning, and sparse GP VAE).

**Questions:**

1. Can you formally explain why using HiPPO can address the forgetting issue?

2. What is the benefit of using time-varying bases here, when compared with stationary bases?

3. In sec 3, in the integral with basis functions (e.g., line 133, line 146, etc.) what's the domain the integral is defined? Previously in in sec 2.5, the integral is from `-\inf` to `t`, is it also the case here? or from `-\inf` to `\inf`?

4. Can you explain why the matrix ODE version leads to numerical instability?

5. From the current paper, I still don't fully follow how the mult-dimensional input case is derived. How is the integral originally defined on the real axis extended to here?

**Ethical Concerns:**

["NO or VERY MINOR ethics concerns only"]

**Final Justification:**

The rebuttal addressed my concern especially about the adaptivity.

**Limitations:**

No. There's no explicit sections or paragraphs on limitation, although some limitations are mentioned in the writing. I suggest to make them more obvious.

**Quality:**

3

**Strengths And Weaknesses:**

Strengths

1. The combination of GP and HiPPO to my understanding is novel
2. The overall structure and the logic is clear (though some details are missing see Questions).
3. The authors conducted diverse experiments to validate the proposed approach.

Weakness.

1. It is unclear from the theoretical side the benefits of using basis functions in HiPPO in the online setting. The current paper merely draws an analogy to the time series approximation in RNN.

2. The paper is motivated by updating inducing points may lead to forgetting. But it doesn't give a formal theoretical argument why using the HiPPO bases can avoid this issue.

2. It is unclear whether the proposed algorithm is scalable to high-dim, complex problems with large data. The experimental results are mainly low-dimensional. However, I acknowledge this is a quite prevalent limitation of GPs in general.

---

> ### Author Rebuttal · Authors · 2025-07-30
>
> ### Global Response:
>
> We sincerely thank all reviewers for the constructive feedbacks and the overall positive reviews. We are encouraged that reviewers appreciated our **novel (CC7b, GFdF, Aht3)** and **technically sound (D83n, GFdF)** method that addresses the **critical (D83n)** catastrophic forgetting problem in online GPs. In general, the reviewers are happy with the **clarity (D83n, Aht3)** of the manuscript and our **comprehensive (CC7b, HjVk) experimental evaluation** of OHSVGP, which shows **strong performance (D83n, HjVk, GFdF, Aht3)** compared with baselines.
>
> ---
> ### Individual Response to Reviewer CC7b:
>
> Below we address the questions from Reviewer CC7b.
>
> > Q1. Benefit of HiPPO basis in online setting and why it can address the forgetting issue.
>
> One of the key benefits of HiPPO basis is its adaptive nature, and to our knowledge, OHSVGP is the first work that considers adaptive basis for interdomain GP. Our motivation for this adaptive basis is exactly trying to address the forgetting issue: the basis function solely focus on history, and we discuss it in details in our response to Q3.
>
> Moreover, in conventional OSVGP, when new task data arrives, updating the inducing points by optimizing online ELBO cannot gaurantee there are sufficient inducing points left in the previous task regions as shown in our experiments, which makes the prediction in these past regions close to the uninformative prior since each inducing point can only capture local behavior of GP approximation. In contrast, OHSVGP defines inducing variables as time-dependent orthogonal polynomial basis functions which by design can capture the global functional behavior in all past data regions. In particular, the basis of HiPPO-LegS is constructed based on uniform measure over the whole history which imposes strong inductive bias in the GP approximation to ensure the posterior predictive is equally well in all the past regions (see Appendix E.4 for the comparison between OHSVGP based on HiPPO-LegS and other HiPPO variants).
>
> From a theoretical persepective, given regularity assumption of the function to memorize, its approximation based on orthogonal polynomials basis in HiPPO frameworks are theoretically shown to be asymptotically exact in L2 sense with $M=\infty$ (Proposition 6 in [1]). Although it is a result in the limit, this asymptotically perfect memory property does not hold for any arbitrary basis functions (i.e., the memory constructed based on many other basis functions are gauranteed to be lossy even in the limit). Empirically, HiPPO based RNNs and the following SSMs (S4, Mamba) inspired by it have shown profound success in sequence modeling tasks requiring long-term memory.
>
> [1] "HiPPO: Recurrent Memory with Optimal Polynomial Projections", NeurIPS 2020
>
> > Q2. Scalability to high-dimensional problems.
>
> We conduct the experiments with relevant baselines on commonly used datasets in the context of GPs. Based on the overall positive results, we believe our method is one of the state-of-the-art GP based methods in online/continual learning. Indeed, GPs may have issues when scaling up to high-dimensional inputs. This is an interesting and open research question of generic GP or kernel methods orthogonal to the contribution of this work, which is about building reliable long-term memory.
>
> > Q3. Benefit of time-varing basis against stationary ones.
>
> The "time-varying" nature of HiPPO bases is a principled way to handle the expanding time domain in online learning. Specifically, at time $t$, the orthogonal polynomial basis functions are defined on $[0,t]$. As new data arrives, this interval expands to $[0,t+Δt]$ to cover the new time region where the new data live, and the basis functions adapt accordingly. This is fundamentally different from stationary bases on a fixed interval. The key benefits are:
> * Adaptive coverage: The bases always span exactly the observed history, no more, no less. An unnecessarily large coverage including regions where no data exists may lead to underfitting in the historical region (see the comparison between OVFF and OHSVGP in Sec 5.1), and a small coverage excluding some regions with data will make the prediction in those regions close to the uninformative prior. The issue of suboptimal coverage has also been discussed in previous works such as [2].
> * No need for pre-specification: Unlike stationary bases, we don't need to know the final time horizon covering all the future tasks in advance, which is in general impossible for real world online learning problems.
>
> [2] "Variational Fourier Features for Gaussian Processe", JMLR 2018
>
>
> > Q4. Domain of the integral in Sec 3.
>
> The domain of the integral for generic HiPPO is from $-\infty$ to $t$ as in Sec 2.5. For HiPPO-LegS, since the measure it uses is uniform over [0, t], $\omega^{(t)}(x)=\frac{1}{t}\mathbb{1}_{[0,t]}(x)$, the domain of the integral essentially becomes from $0$ to $t$ by noticing that the time varing basis is defined as $\phi^{(t)}(x)=g^{(t)}(x) \omega^{(t)}(x)$.
>
> > Q5. Why the matrix ODE version of $K_{uu}^{(t)}$ leads to numerical instability.
>
> The instability of the direct ODE approach compared with RFF approach can be seen from the difference in the forms of their evolutions (especially the first term):
>
> * RFF approach: The evolution of Fourier fearure is of the form $\frac{d}{d t} \mathbf{Z}_w^{(t)}=A(t) \mathbf{Z}_w^{(t)}+\cdots$, which includes evolving vectors with the operator $\mathcal{L}_1: X \mapsto A(t) X$.
>
> * Direct ODE approach: The direct evolution of $K_{uu}^{(t)}$ is of the form $\frac{d}{dt} K_{uu}^{(t)} = [A(t) K_{u u}^{(t)}+K_{u u}^{(t)} A(t)^{\top}] + \cdots$, which requires the Lyapunov operator $\mathcal{L}_2: X \mapsto A(t) X+X A(t)^{\top}$.
>
> The critical difference is that $\mathcal{L}_2$ has eigenvalues $\lambda_i+\lambda_j$ (where $\lambda_i, \lambda_j$ are eigenvalues of $A(t)$), while $\mathcal{L}_1$ has eigenvalues $\lambda_i$. Since HiPPO-LegS uses a lower-triangular $A(t)$ with negative diagonal entries, the eigenvalues $\lambda_i<0$. The Lyapunov operator can thus have eigenvalues $\lambda_i+\lambda_j$ that are approximately twice as negative, leading to a stiff ODE system with poorer numerical conditioning.
>
> While we find simple numerical methods, such as Euler method may be unstable to solve this ODE, we will explore fast and numerically stable solvers for this matrix ODE in future work.
>
> > Q6. How is the mult-dimensional input case is derived
>
> Given an ordered training batch $\\{x_1, \cdots, x_N\\}$, Eq. 5 can be viewed as a discretization (with step size $\Delta t$) of an ODE solving a path integral $\int k(x_n, x(s)) \phi^{(t)}(s) ds$. The $i$-th training input $x_i$ is assumed to be $x_i:=x(i \Delta t)$ and thus the path integral is approximately solved with discretized recurrence based on the training inputs corresponding to $\\{x(i\Delta t)\\}_{i=1}^N$. Notice that different orders of the training batch implicitly define different path integrals to approximate. While they all define valid interdomain inducing variables, the order does matter for the performance of OHSVGP in multi-dimensional input continual learning as demonstrated in our UCI experiments and the 2D-input visualization example in Appendix E.3.

---

> > ### Comment · Reviewer_CC7b · 2025-08-05
> >
> > Thanks for the rebuttal. They helped solve my confusion. I raised my score.

---

### Note · Authors · 2025-08-13

We would like to take this opportunity to thank all reviewers again for their thoughtful feedback and positive evaluations. We are excited about our HiPPO based recurrent memory for online learning with sparse Gaussian processes, which to our knowledge, is the first interdomain GP method based on adaptive basis functions that are naturally suitable for online learning tasks. OHSVGP consistently demonstrates remarkable long range memory preservation ability across various online or continual learning tasks. We therefore believe our work will be of high interest to the online/continual learning, and probabilistic machine learning research communities at NeurIPS. Moreover, we believe that all questions have been addressed during the rebuttal period, and if accepted, we will incorporate the discussions from the rebuttal into the revised manuscript.

---

### Decision · Program_Chairs · 2025-09-17

**Decision:**

Accept (poster)

**Comment:**

The authors propose OHSVGP, a new online Gaussian Process model that leverages HiPPO to efficiently retain past information and update as new data arrives. Reviewers found the work to be both novel and elegant—particularly the use of HiPPO projection coefficients as time-evolving inducing variables—and effective based on the experimental results. The paper was also considered well presented, though I encourage the authors to further refine the presentation based on the reviewers’ feedback and discussions.